# Sensory Schwann cells set perceptual thresholds for touch and selectively regulate mechanical nociception

Julia Ojeda-Alonso[1,9], Laura Calvo-Enrique [2,8,9], Ricardo Paricio-Montesinos [3,4], Rakesh Kumar [2,5], Ming-Dong Zhang [2], James F. A. Poulet [3,6], Patrik Ernfors [2] ✉ & Gary R. Lewin [1,6,7] ✉

Previous work identified nociceptive Schwann cells that can initiate pain. Consistent with the existence of inherently mechanosensitive sensory Schwann cells, we found that in mice, the mechanosensory function of almost all nociceptors, including those signaling fast pain, were dependent on sensory Schwann cells. In polymodal nociceptors, sensory Schwann cells signal mechanical, but not cold or heat pain. Terminal Schwann cells also surround mechanoreceptor nerve-endings within the Meissner's corpuscle and in hair follicle lanceolate endings that both signal vibrotactile touch. Within Meissner´s corpuscles, two molecularly and functionally distinct sensory Schwann cells positive for Sox10 and Sox2 differentially modulate rapidly adapting mechanoreceptor function. Using optogenetics we show that Meissner's corpuscle Schwann cells are necessary for the perception of low threshold vibrotactile stimuli. These results show that sensory Schwann cells within diverse glio-neural mechanosensory end-organs are sensors for mechanical pain as well as necessary for touch perception.

Touch and pain sensations are conveyed by sensory neurons with their cell bodies located in the dorsal root ganglia (DRG). DRG neurons can be broadly classified as mechanoreceptors responsible for touch sensation or nociceptors which detect harmful mechanical, thermal and chemical stimuli[1]. The last three decades have seen huge advances in the discovery of nociceptor-specific ion channels and other signaling molecules that may be targeted to control pain[2–4]. More recently molecules necessary for normal mammalian touch sensation have also been identified[5–7]. Such studies were predicated on the idea that the transduction of mechanical stimuli takes place primarily at the sensory neuron membrane. Consistent with this idea many studies have shown

that cultured sensory neurons, whether mechanoreceptors or nociceptors, are mechanosensitive and mechanical stimuli activate fast inward currents in these cells[5,7–12]. However, Merkel cells in the touch dome complex that are innervated by type I slowly-adapting mechanoreceptors (SAM) are also known to be mechanosensitive[13,14]. Furthermore, optogenetic activation of Merkel cells drives action potential firing in mechanoreceptors through a mechanism that is thought to involve release of transmitter substances[15]. Keratinocytes are another skin cell that have proposed to be involved in modulating the mechanosensitivity of nociceptors[16–18]. However, in the case of keratinocytes and Merkel cells there is no indication that these cell

[1]Molecular Physiology of Somatic Sensation, Max Delbrück Center for Molecular Medicine, 13125 Berlin, Germany. [2]Department of Medical Biochemistry and Biophysics, Division of Molecular Neurobiology, Karolinska Institutet, Stockholm, Sweden. [3]Neural Circuits and Behavior, Max Delbrück Center for Molecular Medicine, 13125 Berlin, Germany. [4]Deutsches Zentrum für Neurodegenerative Erkrankungen e. V. (DZNE), Venusberg-Campus 1/99, 53127 Bonn, Germany. [5]Pain Center, Department of Anesthesiology Washington University School of Medicine, CB 8108, 660 S. Euclid Ave., St. Louis, MO 63110, USA. [6]Charité-Universitätsmedizin Berlin, Charitéplatz 1, 10117 Berlin, Germany. [7]German Center for Mental Health (DZPG), partner site Berlin, Berlin, Germany. [8]Present address: Departamento de Biología Celular y Patología, Instituto de Neurociencias de Castilla y León, University of Salamanca, Salamanca, Spain. [9]These authors contributed equally: Julia Ojeda-Alonso, Laura Calvo-Enrique. ✉e-mail: patrik.ernfors@ki.se; glewin@mdc-berlin.de

types are essential for initiating fast mechanosensory responses in primary sensory neurons, i.e. responses in the millisecond range. We recently identified a new cell type which we have termed nociceptive Schwann cells that express the transcription factor Sox10. Many Sox10[+] cells are tightly associated with nociceptor sensory endings and their excitation can initiate pain[19,20]. We have also identified Sox10[+] and Sox2[+] Schwann cells that are closely associated with mechanoreceptor endings in skin end-organs needed for touch sensation[5,19,21,22]. The role of these cells in the transduction of light touch or vibration, however, has remained unexplored. Here, we used optogenetic tools to directly assess the contribution of Sox10[+] and Sox2[+] sensory Schwann cells to the transduction of mechanical signals by nociceptors and their roles in the perception of touch.

## Results

### Optical activation of Sox10[+] Schwann cells rapidly modulates nociceptors

We generated mice expressing channelrhodopsin (ChR2) or Archaerhodopsin-3 (ArchT) in Sox10 and Sox2-positive terminal Schwann cells to excite or inhibit these cells with light. Light stimulation of nociceptive Schwann cells provokes and modulates nocifensive behaviors[5], and blue light stimulation of nociceptive Schwann cells from Sox10-ChR2 mice evoked fast and sustained inward currents (Supplementary Fig. 1a, b). Nociceptive Schwann cells are innervated by both thinly myelinated (Aδ-fiber) and unmyelinated (C-fiber) sensory axons which have diverse receptor properties[1]. Classically, many C-fiber nociceptors are polymodal responding to both mechanical and thermal stimuli[1,23–25], although polymodal nociceptors can also contribute to non-noxious cool and warm perception[24,26]. Using an ex vivo preparation[5,25], we recorded from identified nociceptors to determine the contribution of sensory Schwann cells to nociceptor function (Fig. 1a). In Sox10-ChR2 mice, blue light initiated sustained action potential firing in all four types of nociceptors recorded: Aδ-mechanonociceptors (A-Ms), C-mechanonociceptors (C-M responding only to mechanical stimuli), polymodal nociceptors, including C-MH (C-mechanoheat), C-MC (C-mechanocold) or C-MHC (C-mechanoheatcold) nociceptors that respond to mechanical, cold, heat or both[24,27,28], and C-thermoreceptors (responding to thermal, but not mechanical stimuli) (Fig. 1b–e, Fig. 2a–c; Supplementary Fig. 1). All types of nociceptors were maximally activated with light intensities of 2.6 mW/mm² and above (Supplementary Fig. 1c). Around 50% of A-Ms and C-M fibers tested were excited by blue light, but the tonic firing responses of these neurons was substantially lower compared to that evoked by supramaximal mechanical stimuli (Fig. 1f, g). In contrast, almost all polymodal nociceptors were robustly driven by blue light with similar firing frequencies to those evoked by supramaximal mechanical stimuli (Fig. 1d, i). Interestingly, light-evoked activity in both C-Ms and polymodal nociceptors showed extremely short latencies, probably reflecting particularly tight coupling between the nociceptive Schwann cell and C-fiber ending (Fig. 1e). Indeed, careful examination of individual neuronal latencies to blue light revealed that the vast majority fired with delays of much <100 ms after light onset (Fig. 1e). In contrast, the latencies for mechanical stimuli were substantially and significantly longer than those found with light stimuli for both C-M and C-polymodal nociceptors (Fig. 1e). Longer mechanical latencies were due to the fact that the mechanical probe driven by the piezoelectric device moves at a finite velocity and it takes some time (mostly >50 ms) before the probe exerts sufficient force to excite the high threshold nociceptor.

To test the contribution of nociceptive Schwann cells to endogenous mechanosensitivity we compared the response of nociceptors to a supramaximal mechanical stimulus and thermal stimuli before and after a 10-min exposure to yellow light in Sox10-ArchT mice compared to Sox10-Cre mice, lacking ArchT (Fig. 1l). We chose this experimental design for two reasons. First, in behavioral experiments we had used

30 min of yellow light which was sufficient to alter pain behaviors[19]. Second, this design allowed us to make statistically robust comparisons between the mechanosensitivity of single neurons exposed to yellow light when ArchT was present in Schwann cells or not. This was important as in order to test the effects of optogenetic manipulation on primary afferent responses it is necessary to mechanically stimulate the receptive fields repeatedly with suprathreshold stimuli. It is well known that nociceptors in particular can display sensitization or desensitization following repeated noxious stimulation[27,29,30]. Thus, our protocol allowed us to control for the effects of repeated stimuli which could be mistaken for effects of light exposure. Of the A-M fibers subjected to light, 60% (6/10 fibers) displayed an elevation in threshold and reduction in mechanically evoked activity (here defined as >20% reduction) (Fig. 1i, j, Supplementary Fig. 1j). In comparison, none of the A-Ms recorded from mice lacking ArchT expression (N = 6) displayed any change in threshold or mechanically evoked activity during the same period and using the same stimuli (Fig. 1j, Supplementary Fig. 1j). There was a >50% reduction in mechanosensitivity in AMs recorded from Sox10-ArchT mice which was significantly different from controls (AMs from Sox10-Cre mice) at 10 and 20 min following yellow light, Two-way ANOVA, P < 0.020, Bonferroni's multiple comparison test (Fig. 1j). We also examined all C-fibers with a mechanosensitive receptive field (C-Ms and polymodal C-fibers). More than 78% (N = 26/33) of these afferents showed a >20% reduction in mechanical evoked activity after yellow light exposure in Sox10-ArchT mice. However, C-fibers recorded from Sox10-Cre controls showed no increase in threshold or reduction in response to suprathreshold stimuli following yellow light exposure (N = 10) (Fig. 1k, l, Supplementary Fig. 1k). Of the C-M fibers exposed to yellow light 68% (13/19) showed a >20% decrease in mechanically evoked firing after yellow light exposure, a higher proportion of C-Ms than were robustly excited by blue light. The mean mechanical thresholds for C-M activation also rose considerably after yellow light, but this change was not statistically different from controls recorded from Sox10-Cre mice (Fig. 1k, Supplementary Fig. 1k). In Sox10-ArchT mice almost all C-polymodal fibers (13/14) showed a robust decrease in mechanosensitivity following yellow light. The decreased sensitivity of both C-M and C-polymodal nociceptors was immediately apparent after the end of the yellow light stimulation and persisted 10 and 20 min after light exposure, and this was statistically significant compared to controls (C-fibers from Sox10-Cre mice) Two-way ANOVA, P < 0.001, Bonferroni's multiple comparison test (Fig. 2k, l, Supplementary Fig. 1k). Yellow light exposure was associated with a slight increase in the threshold of C-polymodal fibers from Sox10-ArchT mice, but the same effect was seen in C-fibers from control Sox10-Cre mice, thus elevated C-fiber thresholds probably reflects mild stimulus evoked desensitization (Fig. 1l).

### Schwann cells specifically control mechanosensitivity, but not thermal sensitivity

C-fibers that only responded to thermal stimuli were all strongly driven by blue light in Sox10-ChR2 mice (N = 7), albeit with longer latencies than mechanosensitive C-fibers (Fig. 2a–c, Supplementary 2a, b). Thus, there was strong connectivity between nociceptive Sensory Schwann cells and thermosensitive C-fibers. To assess whether this connectivity contributes to thermal sensation we silenced Schwann cells and quantified thermally induced nerve activity. Both polymodal and C-thermoreceptors respond to cooling or heating of the skin. We thus quantified thermally evoked activity in both these C-fiber types recorded from Sox10-ArchT (N = 28 C-fibers) and control Sox10-Cre mice (N = 16 C-fibers) before and after yellow light. First, we analyzed the responses of all C-fibers (thermal only and polymodal) with a response either to cooling or heating separately (Fig. 2d–i) and observed that there was no significant change in thermal threshold or thermally evoked spikes in Sox10-ArchT mice at any point after the yellow light compared to Sox10-Cre control mice (Fig. 2d–i,

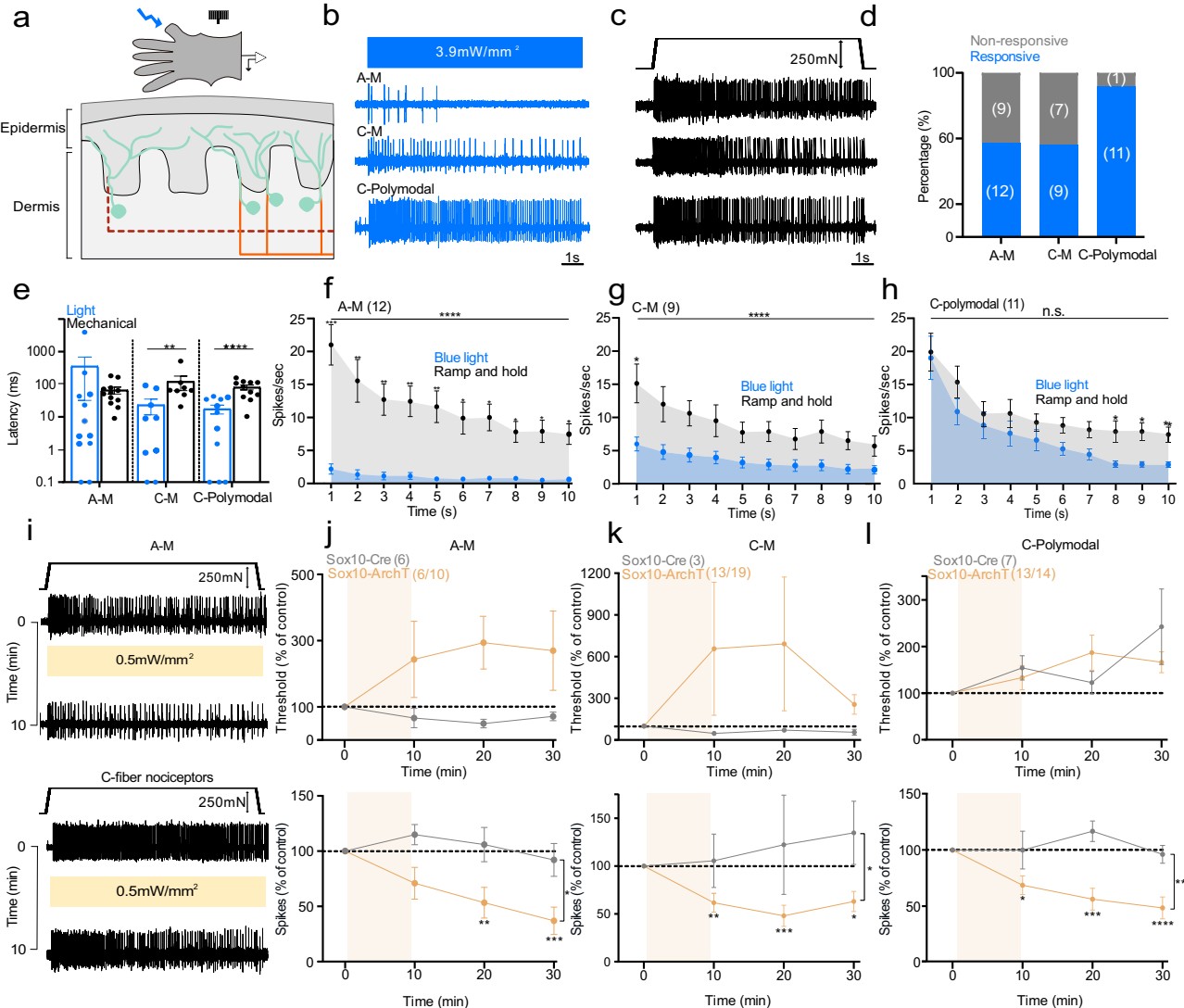

**Fig. 1 | Nociceptive Sox10⁺ Schwann cells are required for nociceptor mechanosensitivity. a** Schematic diagram of ex vivo preparation used to stimulate nociceptive Schwann cells with light. **b, c** Example traces of nociceptor types recorded. **b** Activity was recorded after blue light stimulation. **c** Spiking of the same nociceptor as in **b** to mechanical ramp and hold stimulation (10 s). **d** Proportion of nociceptors responding to blue light. **e** Latency of response to blue light compared to mechanical ramp and hold stimuli (AMs $n = 12$, C-M $n = 9$, C-polymodal $n = 11$) *$P = 0.015$, ***$P = 0.0052$, Mann–Whitney two tailed U-test. **f–h** Mean time course of nociceptor activation (1 s bins, 10 s, 250 mN amplitude ramp and hold stimulus), A-Mechano-nociceptors (AM $n = 12$) (**f**), C-mechano-nociceptors (C-M $n = 9$) (**g**), and polymodal C-fibers with thermal and mechanosensitivity ($n = 11$) (**h**). Mean spiking rates of the same receptors to blue light (blue) and mechanical stimuli (gray).

****$P < 0.001$ two-way ANOVA. **i** Representative traces show A-M and C-fiber mechanonociceptor activity to mechanical stimuli 10 s (ramp and hold) before and after 10 min of yellow light (**i**). **j** A-M-nociceptors threshold (upper panel) and spiking rates (lower panel) to mechanical stimuli before and after yellow light in Sox10-ArchT mice ($n = 6$) and in Sox10-Cre control ($n = 6$) animals (two-way ANOVA, $P = 0.020$, Bonferroni's multiple comparison test). **k** C-mechano nociceptors (C-M) threshold and spiking rates to mechanical stimuli before and after yellow light in Sox10-ArchT ($n = 13$) and control mice ($n = 3$). **l** Polymodal C-fibers response to mechanical stimuli before and after yellow light in Sox10-ArchT ($n = 13$) and control ($n = 7$) mice (two-way ANOVA, $P < 0.001$, Bonferroni's multiple comparison test). Data are presented as mean values ± s.e.m. Source data are provided as a Source Data file.

Supplementary Fig 2c–k). Because mechanosensitive polymodal fibers were included, these data indicated that the transduction of thermal stimuli was unchanged in the same C-fibers that showed substantial reductions in mechanically evoked activity (Fig. 1l). When the thermal response of all types of afferents were analysed separately during yellow light stimulation no consistent change in sensitivity was seen in any sub-type (Supplementary Fig. 2d–k). Thus, functionally coupled nociceptive Schwann cells appear to be selectively involved in the transduction of mechanical stimuli with thermal sensitivity likely transduced by ion channels located in the nociceptor membrane. Furthermore, the ineffectiveness of yellow light on thermal responses shows that Schwann cell hyperpolarization has no effect by itself on the electrical excitability of the C-fiber ending.

In summary, a very large proportion of nociceptors of all types depend on nociceptive Schwann cells for normal mechanosensitivity. About half of the mechanonociceptors did not show any functional connectivity with Schwann cells and this could in principle be a technical issue due to incomplete recombination after tamoxifen injections. However, connectivity was much lower in C-M and AM fibers compared to polymodal fibers which suggested that non-responsive nociceptors are physiologically distinct. However, blue light responsive and non-responsive C-M and A-M fibers showed similar mean mechanical thresholds and responses to suprathreshold mechanical stimuli (Supplementary Fig. 1d–f). We carried out a similar analysis on A-M and C-fiber nociceptors that were either inhibited or not by yellow light in Sox10-ArchT mice. Here again there were no clear differences

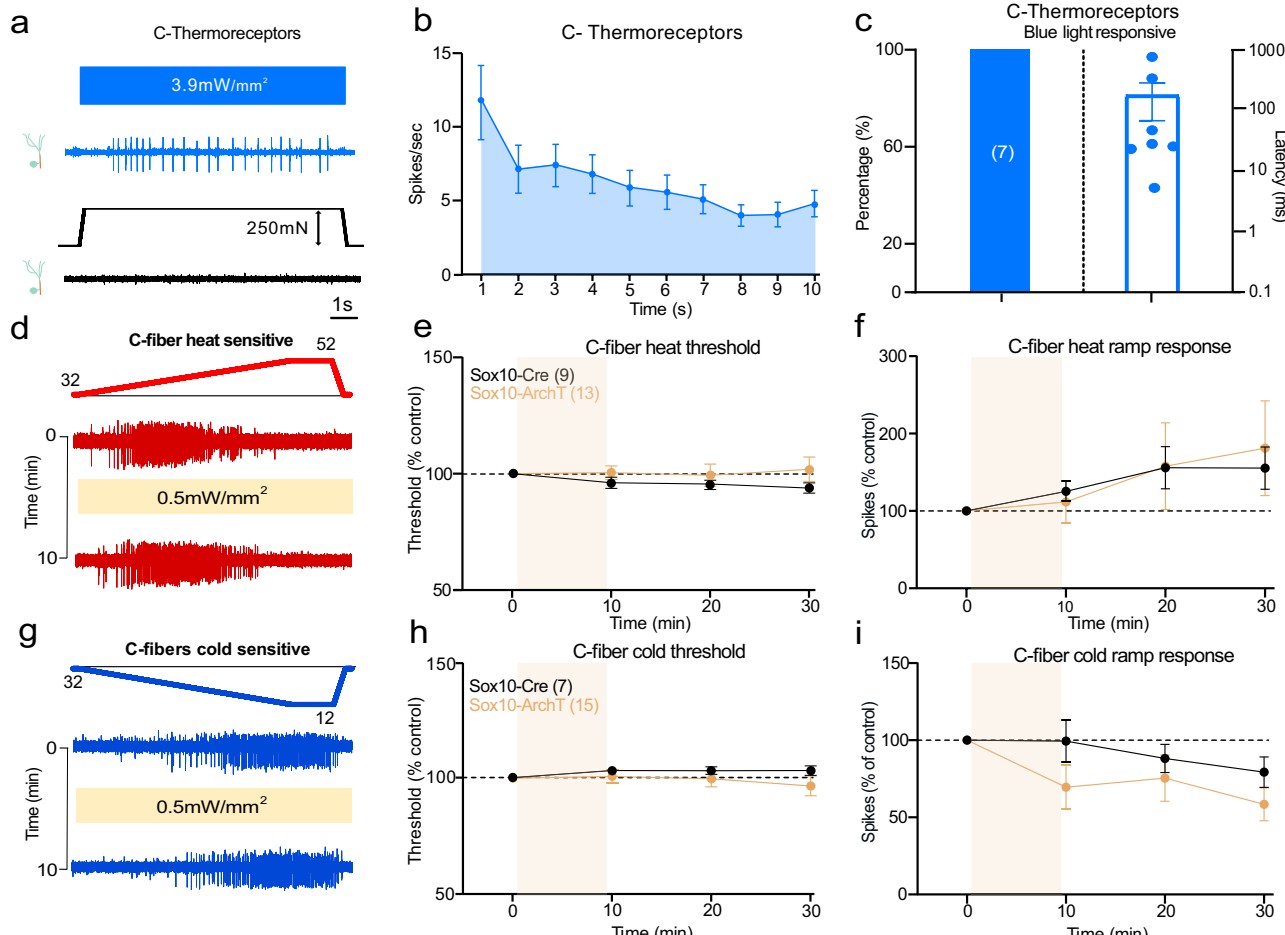

**Fig. 2 | Nociceptive Sox10+ Schwann cells are not required for thermal sensitivity. a** Example of single-unit activity in a C-thermoreceptor driven by blue light stimulation (10 s), the same unit did not respond to mechanical stimulation of its receptive field. **b** Mean rate of firing to blue light stimulation of C-thermoreceptors ($n = 7$ units). **c** Proportion of C-thermoreceptors with a blue light response (left) and their latencies (right). **d** Examples traces of C-fibers responding to a ramp heating stimulus before and after yellow light exposure in Sox10-ArchT mice. **e, f** Heat activated responses measured as thresholds (**e**) and spiking rates (**f**) in all heat responsive fibers (C-thermoreceptors and polymodal C-fibers pooled) before and after yellow light in Sox10-ArchT mice ($n = 13$) compared to those from Sox10-Cre control animals ($n = 9$) did not differ (two-way ANOVA, $P = 0.55$ (threshold) and $P = 0.92$ (ramp response), Bonferroni's multiple comparison test). (**g**) Examples traces of C-fibers responding to a ramp cooling stimulus before and after yellow light exposure in Sox10-ArchT mice. **h, i** Cold activated responses measured as thresholds (**h**) and spiking rates (**i**) in all cold responsive fibers (C-thermoreceptors and polymodal C-fibers pooled) before and after yellow light in Sox10-ArchT mice ($n = 15$) compared to those from Sox10-Cre control animals ($n = 7$) did not differ (two-way ANOVA, $P = 0.6$ (threshold) and $P = 0.48$ (ramp response), Bonferroni's multiple comparison test). Data are presented as mean values ± s.e.m. Source data are provided as a Source Data file.

in the mechanical thresholds or suprathreshold responses of nociceptors that were inhibited or not by yellow light. (Supplementary Fig. 1g–i).

### Sox10+ Schwann cells are functionally coupled to low threshold mechanoreceptors

Sox10-TOM-labeled cells were also found within Meissner's corpuscles (Fig. 3a, b) and hair follicles[19]. Consistent with Abdo, et al.[19], Sox10-TOM labeling was not observed in any cutaneous afferents (Fig. 3a, b, Supplementary Fig. 3). In most Meissner's corpuscles we found 2–4 Sox10-TOM+ cells to be intimately associated with the sensory endings of rapidly adapting mechanoreceptors (RAMs), that are required for fine touch perception in mice and humans[5,7,21,22,31] (Supplementary Fig. 3). We used blue light to selectively activate Schwann cells in the glabrous or hairy skin whilst making single-unit recordings from RAMs (Aβ-fibers innervating Meissner's corpuscles, or Aβ-fibers innervating hair follicles). In Sox10-ChR2 mice blue light reliably evoked 1–2 ultra-short latency spikes in 35% of the glabrous skin RAMs (6/17), but only activated 15% of RAMs in hairy skin (3/20) (Fig. 3c, Supplementary Fig. 4a, b). SAMs were never activated by blue light in the glabrous or hairy skin of Sox10-ChR2

mice (Fig. 3b), consistent with expression of Sox10 in Schwann cells, but not in Merkel cells. To evaluate the contribution of corpuscle resident Schwann cells to mechanosensitivity we compared light-evoked activity with activation by mechanical stimuli in the same neuron (Fig. 3d–f). Ramp and hold mechanical stimuli evoke RAM activity only during the ramp phase of the stimulus as these receptors primarily function as movement sensors[5,25,31]. For glabrous skin RAMs the first spike latencies to blue light stimulation in Sox10-ChR2 mice were significantly faster (mean $2.8 \pm 0.7$ ms) than mechanically evoked spikes ($9.3 \pm 1.8$ ms, $P < 0.01$ unpaired t-test) (Fig. 3c). Blue light activation had an almost instantaneous rise time (Supplementary Fig. 1a, b) whereas the Piezo actuator moved at a finite velocity (15 mm/s), limiting the speed at which the receptor can reach firing threshold. The extremely short latencies for light activation suggest a tight coupling between SOX10+ Schwann cells and the RAM receptor ending. The mechanosensitive properties of RAMs that were unresponsive to blue light were indistinguishable from those activated by blue light in Sox10-ChR2 mice (Supplementary Fig. 4c, d, e, h). There was little indication that blue light significantly altered the mechanosensitivity of RAMs (Supplementary Fig. 4e, f), although this was not tested systematically.

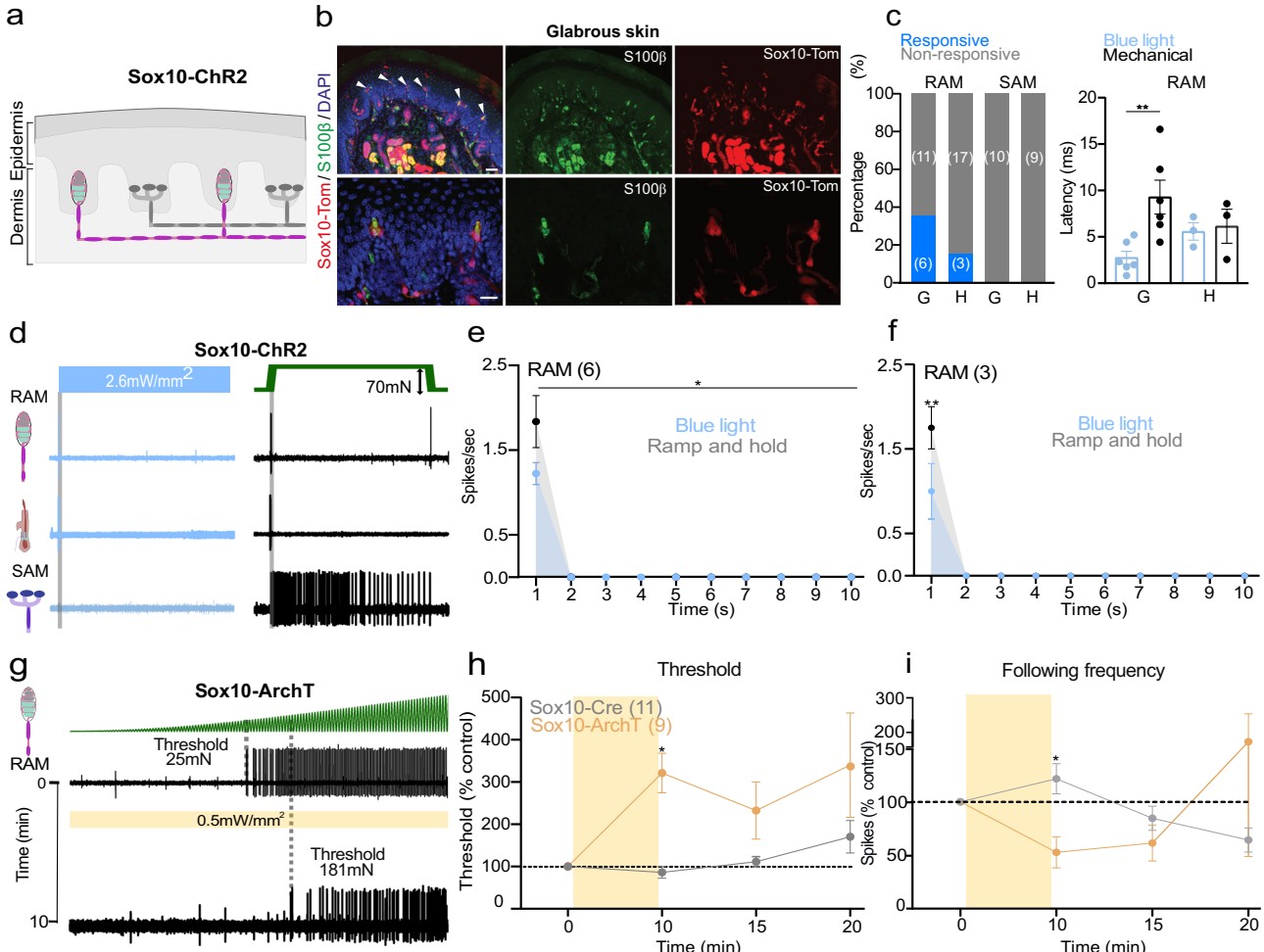

**Fig. 3 | Sensory Sox10+ Schwann cells in the Meissner's corpuscle are required for vibration sensing. a** Schematic diagram of Sox10+ cells in the RAM but not in SAM fibers in the skin. **b** Immunofluorescence showing recombination in glial cells of Meissner corpuscles in Sox10-TOM mice. Immunohistochemistry for TOMATO (recapitulating Sox10 expression) and S100β to label glial cells in the corpuscle sensory ending. Upper panels show a low magnification image of the mice footpads with arrows pointing to Meissner corpuscles. Lower panels show higher magnification images. Scale bar: 50 μm (upper panels) and 20 μm (lower panels). This experiment was repeated at least 6 times. **c** On the left, total number of mechanoreceptors (RAM and SAM) recorded from the hairy (**h**) or glabrous skin (**g**) in Sox10-ChR2 mice, showing proportions of light responsive (blue) and non-responsive mechanoreceptors (gray). On the right, first spike latencies for RAMs comparing optogenetic activation of Schwann cells and mechanical activation of the same afferent during the ramp phase. RAMs recorded from Sox10-ChR2 mice respond faster to light stimulation than to ramp indentation applied at 15 mm/s via a piezo actuator (unpaired t-test, $P = 0.007$). **d** Example of spiking from RAMs and SAMs exposed to blue light compared to a mechanical stimulus recorded from glabrous or hairy skin. **e, f** Mean RAM spiking activity plotted in 1 s bins from Sox10-ChR2 mice during 10 s of blue light or mechanical stimulation from glabrous (**e**) ($n = 6$) or hairy skin (**f**), ($n = 3$). **g** Mechanoreceptor spiking rates in response to 20 Hz vibration stimulus before and after optogenetic inhibition of Schwann cells. Top, RAM representative trace; Bottom, the same unit 10 min after yellow light exposure. **h** Mechanical threshold for first spike for units recorded in Sox10-ArchT ($n = 11$) and control mice ($n = 9$). An increase in the force necessary to evoke the first action potential was observed in RAMs recorded from Sox10-ArchT mice, Two-way ANOVA, $P = 0.0383$, Bonferroni's multiple comparisons test (two sided). **i** The following frequency decreased after yellow light stimulation in Sox10-ArchT+ mice at 10 min and this was statistically significant (Two-way ANOVA, $P = 0.047$, Bonferroni's multiple comparisons test). Data are presented as mean values ± s.e.m. Source data are provided as a Source Data file.

We next used light-induced silencing to ask if Sox10+ Schwann cells within the Meissner corpuscles contribute to the coding of vibrotactile stimuli. In Sox10-ArchT and control Sox10-Cre mice we evaluated RAM sensitivity using a 20 Hz sinusoidal stimulus with a linearly increasing amplitude (Fig. 3g)[22]. We measured the force amplitude for the first spike as well as frequency following (where 1.0 denotes a spike evoked by every sinusoid) before and after 10 min of cyclical yellow light was focused on the receptive field. Again, we classified mechanoreceptors as responsive if threshold was elevated by >20% following yellow light. In Sox10-ArchT mice, almost half of the RAMs (9/20 tested) showed a 3-fold elevation in mean mechanical threshold immediately after the end of the light stimulation compared to control values (Fig. 3g–i) and this was statistically significant (Two-way ANOVA, Bonferroni's multiple comparisons test $P = 0.0383$)

Interestingly, as observed in nociceptors the proportion of receptors exhibiting silencing was higher than the proportion of neurons excited by blue light (45% versus 35%). The mean mechanical threshold of RAMs did not change in control Sox10-Cre mice, Two-way ANOVA, $P = 0.14$, Bonferroni's multiple comparison test (Fig. 3g, h). The number of sinusoid evoked spikes from RAMs in control Sox10-Cre mice remained unchanged after yellow light exposure, but in Sox10-ArchT mice decreased to half of control values immediately after the light stimulus ended and recovered to control levels after 20 min, Two-way ANOVA, $P = 0.09$, Bonferroni's multiple comparison test (Fig. 3i). Note that RAMs unresponsive to yellow light in Sox10-ArchT mice had similar mechanosensitivity to those inhibited by yellow light (Supplementary Fig. 5a–f). SAMs are not associated with SOX10+ Schwann cells and when these receptors were exposed to yellow light we observed no

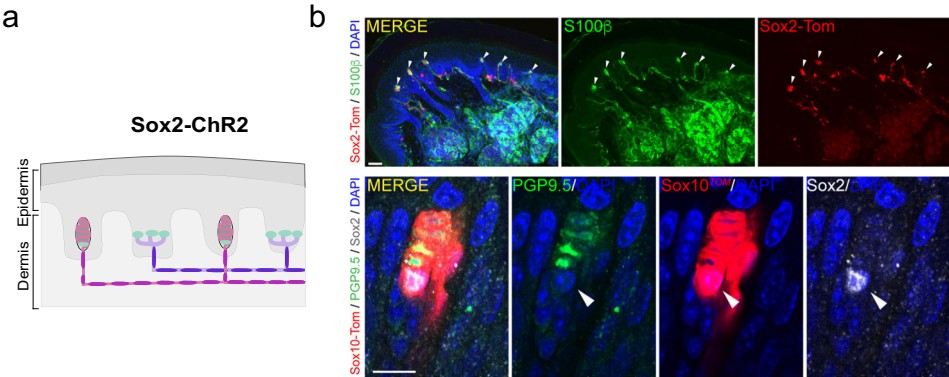

**Fig. 4 | Sox2⁺/Sox10⁺ cells are located at the base of the Meissner corpuscle.**
**a** Schematic of the localization of Sox2 at mechanoreceptor endings. **b** Upper panels show a lower magnification immunohistochemistry for TOMATO (recapitulating Sox2 expression) and S100β to label glial cells in glabrous skin of Sox2-TOM hind paws. Arrowheads point to Meissner corpuscles. Lower panels show a high magnification image of an immunohistochemistry for TOMATO (recapitulating Sox10 expression), PGP9.5 to label neurons in the corpuscle sensory ending and against Sox2. Arrowhead points to a Sox2⁺/Sox10⁺ cell within the Meissner corpuscle. Scale bar: 50 μm (upper panel) and 20 μm (lower panel). Immunohistochemistry experiments were repeated at least 3 times with similar results.

consistent increase in mechanical threshold or number of evoked spikes per mechanical stimulus (Supplementary Fig. 6a–d). Our data thus support the idea that SOX10⁺ Schwann cells within the Meissner corpuscle contribute substantially to setting the threshold and sensitivity of mechanoreceptors required for fine touch.

## Sox2⁺ Schwann cells are functionally coupled to low threshold mechanoreceptors

Using Sox2-TOM reporter mice we noticed that a sub-population of sensory Schwann cells in the Meissner's corpuscles were positive for Sox2. We had previously observed that terminal Schwann cells associated with the lanceolate endings of hair follicles were also Sox2⁺ and that Merkel cells associated with SAMs are Sox2⁺[19]. Here we found just 1–2 Sox2⁺ cells per Meissner's corpuscle (2 Sox2⁺ cells in 3/9 corpuscles, 6/9 had just 1 Sox2⁺ cell) and these cells were preferentially located at the base of the corpuscle (Fig. 4, Supplementary Fig. 3). An overview of the glabrous skin shows that the Sox2-TOM positive cells were specifically found in Meissner's corpuscles and not in other skin cell types, besides Merkel cells (Fig. 4). A closer examination of Sox2⁺ cells in Meissner's corpuscles revealed that these cells were positive for both Sox2 and Sox10 (Fig. 4b, Supplementary Fig. 3). The unusual distribution of Sox2⁺ Schwann cells thus prompted us to examine their function using optogenetics. We first used Sox2-ChR2 mice to examine whether mechanoreceptors could be activated via Sox2 Schwann cells. In Sox2-ChR2 mice we found that between 15–30% of both RAMs and SAMs were activated by blue light in both glabrous and hairy skin (Fig. 5a). In contrast to our findings with Sox10-ChR2 mice, the latencies of activation for all types of mechanoreceptor were uniformly much longer (means >500 ms) than those for the mechanical stimulus used to stimulate the same receptor (Fig. 5b, c). But most strikingly, blue light activation of Sox2-ChR2 cells always evoked low frequency sustained firing from both RAM and SAMs (Fig. 5d–f, Supplementary Fig. 7a–c). Blue light stimulation for 10 s evoked non-adapting firing with a frequency of around 1 Hz in RAMs of the glabrous skin associated with Meissner's corpuscles. Similar low frequency firing responses were observed in light sensitive RAMs of the hairy skin (Supplementary Fig. 7a, b). As expected from previous studies[13,32,33], blue light reliably evoked tonic discharges in ~30% of SAMs tested in Sox2-ChR2 mice consistent with the expression of Sox2 in Merkel cells[19,34]. We noted that the firing rates of SAMs to a 10 s long mechanical stimulus were much higher than those evoked by blue light stimulation, which were similar to those found in RAMs (Fig. 5f). As found for RAMs that were excited by blue light in Sox10-Chr2 mice we found no difference in the mechanosensitivity of blue light responsive

and non-responsive RAMs in Sox2-ChR2 mice (Supplementary Fig. 7d–g). However, we did find that the responses of SAMs to mechanical stimuli that were blue light sensitive was significantly enhanced compared to blue light unresponsive SAMs in Sox2-Chr2 mice (Supplementary Fig 7h, i), suggesting that connectivity with Sox2+ Merkel cells could enhance SAM mechanosensitivity.

It was striking that blue light activation of the Sox2 cells can drive tonic firing in two types of RAM receptor from hairy and glabrous skin. Thus, we next asked if inhibition of Sox2 Schwann cells can alter the mechanosensitivity of Meissner's corpuscle associated RAMs. Using Sox2-ArchT mice we used the same protocol of repeated stimulation with a linearly increasing 20 Hz sinusoid stimulus to measure change in threshold and sensitivity specifically induced by Sox2 cell inhibition. In contrast, to what we had observed in Sox10-ArchT mice we found that the majority (12/14 RAMs) showed a >20% elevation in mechanical threshold after yellow light exposure. Indeed, the proportion of RAMs inhibited by yellow light Sox2-ArchT mice was significantly different from the proportion excited by blue light in Sox2-ChR2 mice (Fisher's exact test $P$-value < 0.0001). Thus, RAMs from Sox2-ArchT mice showed a significant elevation in their mechanical threshold and substantial decrease in their ability to follow the sinusoidal stimulus (Fig. 5h, i), Two-way ANOVA, $P = 0.0002$, Bonferroni's multiple comparison test. The thresholds and mechanosensitivity of the RAMs showed signs of recovery 15 and 20 min after yellow light and were no longer significantly different from RAMs recorded from control mice lacking ArchT from 15 min (Fig. 5h, i). These data suggest that Sox2 cells at the base of the Meissner's corpuscle are functionally distinct from Sox10 cells, and exert a powerful effect in conferring mechanosensitivity to most RAMs innervating the corpuscle.

## Sensory Schwann cells maintain normal perceptual touch threshold

Meissner's corpuscle RAMs are the main afferents required for the detection of the smallest perceptible skin vibrations[22]. We decided to examine the role of Sox10⁺ Schwann cells within the Meissner's corpuscle in regulating the perceptual thresholds of mice in a vibrotactile detection task. We chose to use Sox10-ArchT mice rather than Sox2-ArchT mice as in the latter case yellow light would be predicted to inhibit both RAM and SAMs. Thus, Sox10-ArchT mice enabled us to specifically examine the role of Schwann cells within the Meissner's corpuscle for rapid stimulus detection. We adapted a goal-directed tactile perception task[22] for water-restricted, head-restrained mice in which Sox10-ArchT mice were trained to report a 20 Hz sinusoidal stimulus delivered to the forepaw glabrous skin (Fig. 6a–c). After

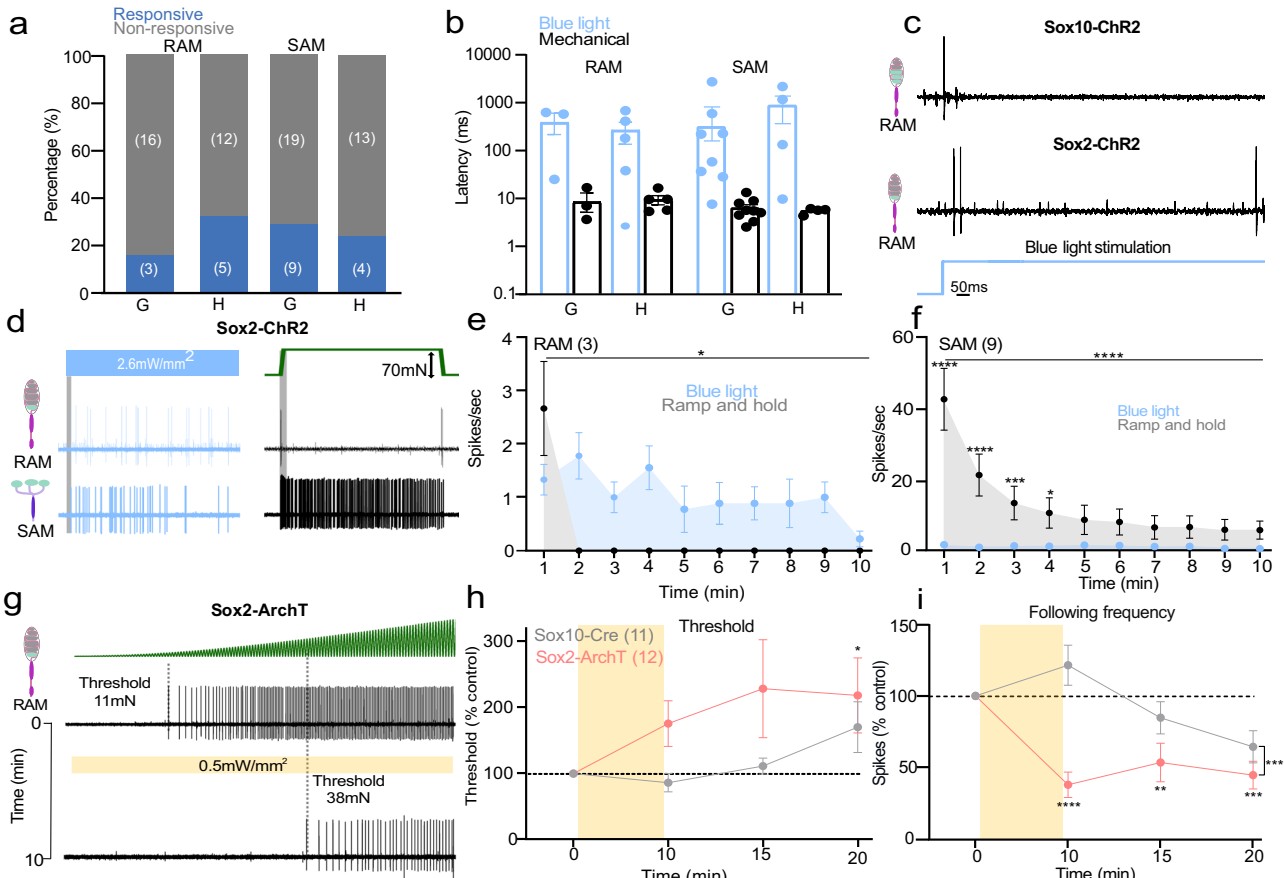

**Fig. 5 | Sensory Sox2+ Schwann cells in the Meissner's corpuscle are required for vibration sensing. a** The proportion of mechanoreceptors (RAM and SAM) recorded from the hairy (H) or glabrous skin (G) in Sox2-ChR2 mice showing activation by blue light (blue), non-responsive (gray). **b** First spike latencies for RAMs comparing optogenetic activation of Schwann cells and mechanical activation of the same afferent during the ramp phase. RAMs recorded from Sox2-ChR2 mice much slower to blue stimulation than to ramp indentation applied at 15 mm/s via a piezo actuator (two sided unpaired *t*-test, P = 0.007). **c** Example of spiking from RAMs exposed to blue light from Sox2-ChR2 mice (top) and Sox2-ChR2 mice (bottom), note long latencies in the latter case. **d** Example traces of spiking from RAMs (top) or SAMs (bottom) exposed to blue light from Sox2-Chr2 mice compared to the response to a mechanical stimulus (right) in glabrous skin. **e** Mean spiking activity plotted in 1 s bins during 10 s of blue light or mechanical stimulation of RAMs in glabrous skin and (**f**) of SAMs in glabrous skin in Sox2-ChR2 mice. **g** Mechanoreceptor spiking rates in response to 20 Hz vibration stimulus before and after optogenetic inhibition of Sox2+ Schwann cells. Top, RAM representative trace; Bottom, the same unit 10 min after yellow light exposure. **h** Mechanical threshold for first spike for Sox2-ArchT and control mice. An increase in the force necessary to evoke the first action potential was observed in RAMs recorded from Sox10-ArchT mice (Two-way ANOVA, P = 0.0383, Bonferroni's multiple comparisons test). **i** The following frequency decreased after yellow light stimulation in Sox2-ArchT+ mice at 10 min and this was statistically significant (two-way ANOVA, P = 0.047, Bonferroni's multiple comparisons test). Data are presented as mean values ± s.e.m. Source data are provided as a Source Data file.

training, mice correctly reported detection of the stimulus by licking a water spout within a time window of 400 ms, the equivalent of 4 sinusoids of the 20 Hz stimulus. The probability of a trained mouse correctly reporting stimulus intensities of 1.5 or 3.0 mN was around 80% P(lick) = 0.8 (Fig. 6d). The next day, we exposed either the forepaw used for behavioral training, or, as a control, the contralateral forepaw, to 30 min of yellow light using the same duty cycle and intensity as used for recordings above. We then retested the mice in the perceptual task immediately after exposure. For the lowest amplitude stimuli used (1.5 or 3.0 mN), mice exposed to yellow light showed a reduction in their ability to correctly detect the stimulus (Fig. 6d). To compare the performance of mice in the detection task we calculated the sensitivity index d′ (see Methods) and found that d′ decreased after yellow light exposure (Fig. 6e). Many of the correct first lick latencies before yellow light were <200 ms, indicating that the mice had perceived the stimulus following only two sinusoids or less (Fig. 6f, Supplementary Fig. 8a–c). First lick latencies increased slightly after the yellow light, but this did not reach statistical significance (Supplementary Fig. 8). Testing on the next day showed that the mice had recovered their perceptual performance back to control levels (Fig. 6g, h).

Importantly, mice exposed to yellow light on the contralateral, untrained, forepaw showed no perceptual deficit (Fig. 6g, h). We went on to test whether the effects of yellow light were due to the presence of ArchT in sensory Schwann cells. In this control experiment, we trained an additional cohort of Sox10-Cre mice that lacked ArchT expression using the same task. As expected, these mice showed no changes to their perceptual threshold following an identical procedure of yellow light exposure of the forepaw (Supplementary Fig. 9). Together, these data indicate that Meissner resident sensory Schwann cell are essential for the mice to perceive vibrotactile stimuli relevant for texture discrimination.

## Discussion

Here we show that Sox10+ Schwann cells form morphologically and functionally diverse glio-neural end-organs. Sox10+ Schwann cells are functionally coupled to both mechanoreceptors and nociceptors and substantially contribute to the mechanosensitivity of both types of receptors (Fig. 7). We also describe a novel function for Sox2+ Schwann cells which represent a sub-population of Sox10+ cells associated with RAMs that innervate hair follicles and Meissner's corpuscles.

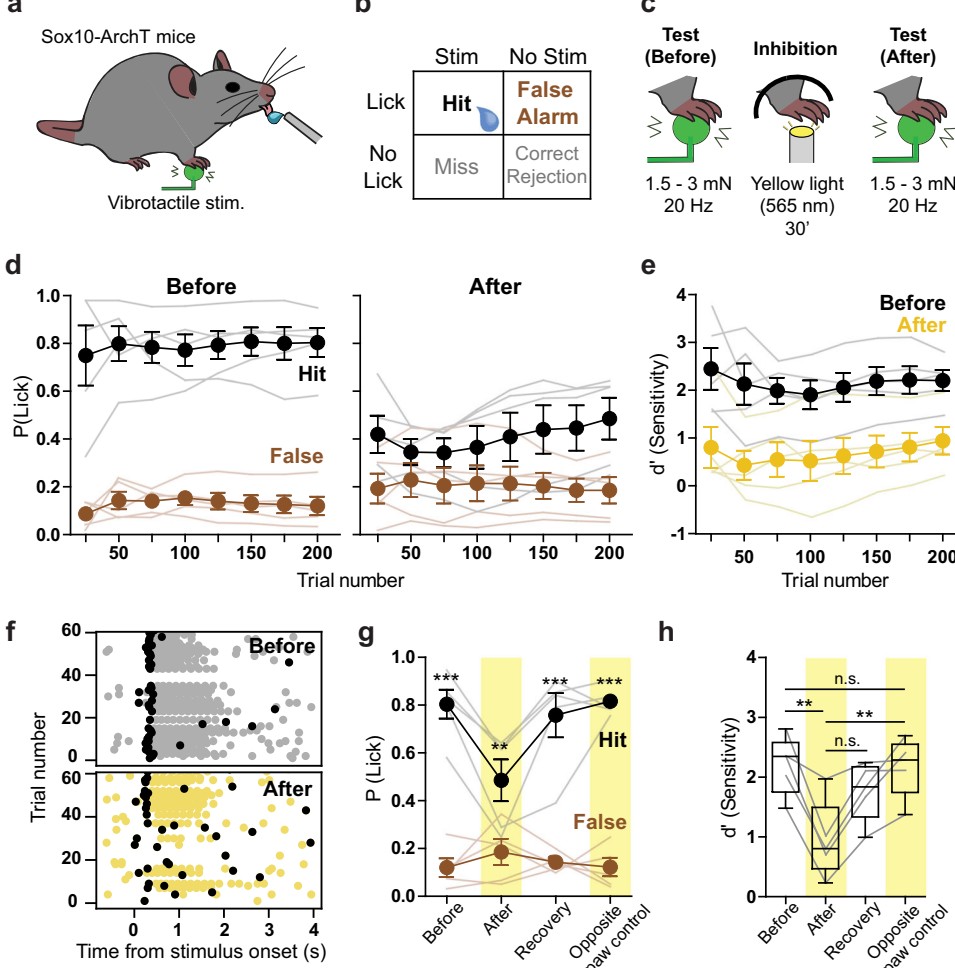

**Fig. 6 | Reduction of vibrotactile perception following optogenetic inhibition of forepaw Sox10+ cells. a** Cartoon schematic showing behavioral setup with right forepaw placed on a vibrotactile stimulator. **b** Structure of the Go/No Go behavioral task. During stimulus trials, mouse licks with a 400 ms window of opportunity following stimulus onset were rewarded and classified as hits. During no stimulus trials, licks in the same window were recorded as false alarms. The proportion of hits and false alarms were later compared to assess performance. **c** Cartoon showing stimulus type and optogenetic stimulation. Mice were trained to report 20 Hz, 1.5 or 3 mN vibrotactile stimuli. Mice that reached a performance level of d′ > 1.5 had their forepaw exposed to yellow light the next day (565 nm, 30′) with the same protocol used in skin-nerve recordings. Immediately after the light exposure, their sensitivity to the same vibrotactile amplitude was again tested. **d** Mean hit and false alarm rates across trials for Sox10-ArchT mice, reporting vibrotactile stimuli on the session before inhibition (left) and after the inhibition (right) (n = 5). **e** Comparison of mean sensitivity index (d′) across trials from same data shown in (**d**) (n = 5). **f** Lick raster plot from an example mouse during vibrotactile detection task, before (top) and after (bottom) the optogenetic inhibition. First lick in each trial is shown in black, other licks in gray or yellow. **g** Session average hit and false alarm rates of Sox10-ArchT mice on different behavior sessions. Statistically significant differences between hits and false alarms were found on the sessions before optogenetic stimulation (before), after optogenetic stimulation (recovery) and after optogenetic stimulation of the tested (after, P = 0.0028, n = 5) and contralateral paw (opposite paw control) (P < 0.001, Two-way ANOVA with Bonferroni post-hoc, n = 5). **h** Session average sensitivity (d′) values of Sox-10-ArchT mice when reporting the vibrotactile stimulus. The sensitivity was lower after the optogenetic inhibition (P = 0.0058) than on the session before. Moreover, the sensitivity was also lower after inhibition of the vibrotactile-sensing paw than after inhibition of the opposite, non-stimulated paw (P = 0.007, two-way ANOVA with Bonferroni post-hoc) (n = 5). Data are presented as mean values ± s.e.m. Box plots show: median at center, upper and lower quartiles at the bounds of box, whiskers are at minima and maxima. Source data are provided as a Source Data file.

Optogenetic silencing of Sox10[+] or Sox2[+] Schwann increased the forces required to activate RAMs. Inhibition of Meissner's corpuscle Sox10[+] Schwann cells associated with RAMs in glabrous skin reversibly increased the perceptual threshold of mice to detect a vibrotactile stimulus. This represents the first direct evidence that Schwann cells within the Meissner's corpuscle are directly involved in the transduction of vibrotactile stimuli, relevant to set perceptual thresholds for touch.

Previous studies on nociceptive Schwann cells showed that optogenetic excitation or inhibition of these cells can initiate and modulate nocifensive behaviors[19]. However, in order to know which types of nociceptors depend on nociceptive Schwann cells it is necessary to directly record from functionally identified nociceptors

with optogenetic manipulations. For example, optogenetic stimulation of keratinocytes can also initiate nocifensive responses in mice which can be attributed to the activation of subsets of Aδ and C-fiber afferents[16]. Blue light activation of keratinocytes evoked extremely long latency responses in nociceptors with typical latencies >10 s[16]. In contrast, blue light excitation of Sox10[+] Schwann cells activated 70% of all nociceptors with latencies much <100 ms and often with latencies of just 1–2 ms. Thus, there appears to be a tight electrical coupling between mechanosensitive nociceptive Schwann cells and the nociceptor ending. Using blue light excitation of nociceptive Schwann cells, we observed that polymodal nociceptors were all as strongly activated by blue light as by a supramaximal mechanical stimulus. This finding is especially striking as existing studies on skin cells like Merkel

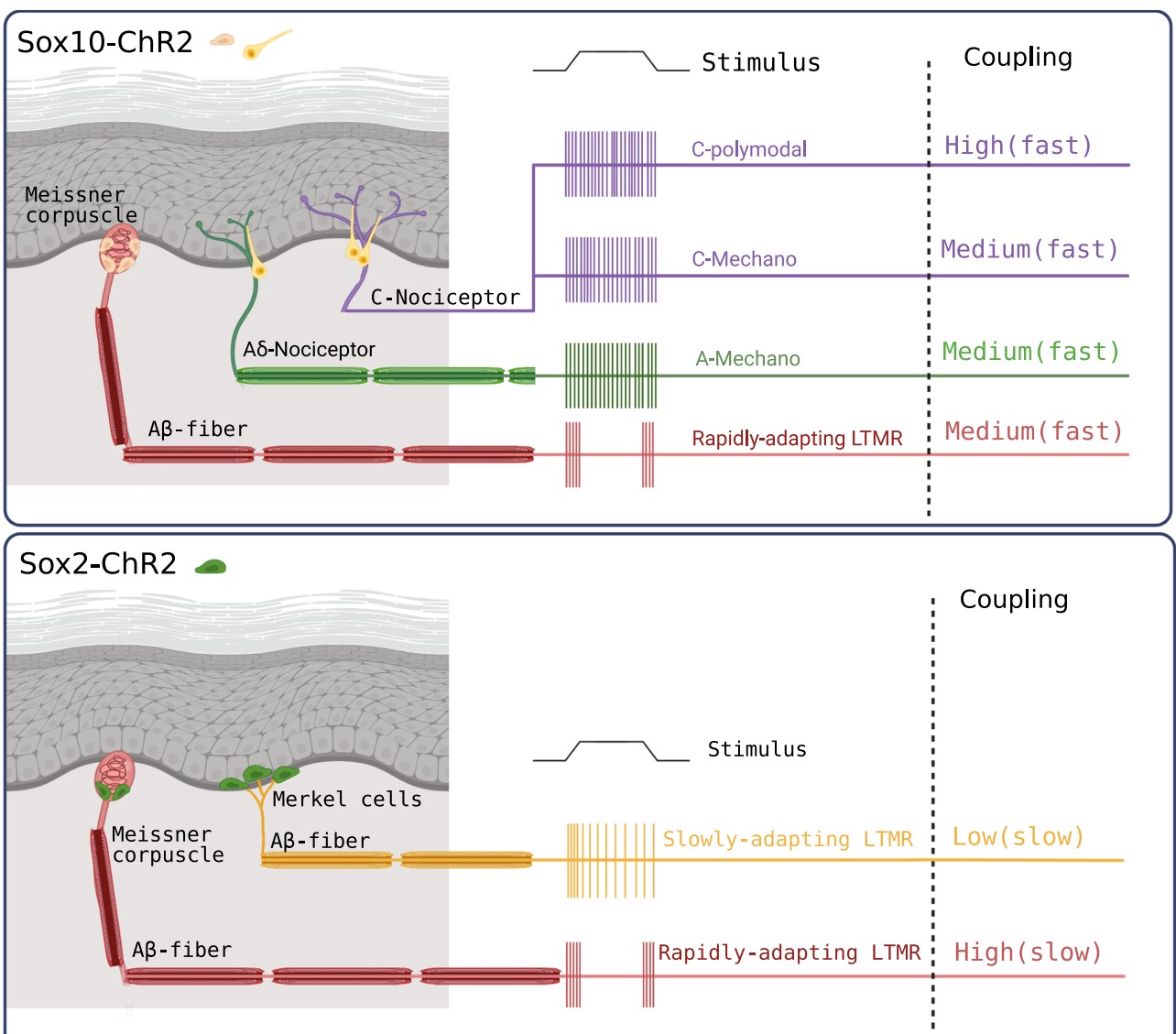

**Fig. 7 | Schematic summary of the functional and morphological diversity of sensory Schwann cells.** Top Sox10 cells are morphologically and functionally diverse. Sox10 cells, in yellow, are found associated with Meissner's corpuscle receptors (left) and show medium coupling (~50% connectivity) that is fast i.e. responses with milliseconds of activation of the Scwann cell. In addition, almost all nociceptor subtypes show medium to very high coupling (~100% coupling in the case of polymodal C-fibers) with Sox10+ cells associated with free nerve endings.

Bottom panel illustrates connectivity of Sox2 cells which are only found associated with low threshold mechanoreceptors (LTMRs). Connectivity was always low (i.e. long latency responses in sensory neurons to optogenetic activation of Sox2 cells). The connectivity was seen in almost all Rapidly adapting LTMRs associated with Meissner's corpuscles and with Slowly-adapting LTMRs associated with Merkel cells. This schematic figure was created with BioRender.com.

cells and keratinocytes have not demonstrated an equivalence between and optogenetically driven responses to those of natural mechanical stimuli[14,16,35,36]. We found that in contrast to polymodal nociceptors, mechanonociceptors (A-M and C-M fibers), some of which are required for fast mechanical pain[37,38], showed less connectivity and much lower firing rates to blue light compared to mechanical stimuli. This specificity is likely due to differential coupling of these two types of nociceptor as it seems unlikely that tamoxifen induced recombination should be more efficient in a subset of Sox10+ cells connected to polymodal nociceptors. Sox10+ Schwann cells which are excitable cells[19] were inhibited by cyclical stimulation of proton pump ArchT for 10 min in our experiment. It is possible that such a prolonged stimulus leads to ionic changes in the extracellular milieu that indirectly inhibit nociceptor ending excitability. For example, protons themselves can powerfully inhibit nociceptor specific voltage-gated sodium channels like $Na_V1.7$[39,40]. However, in polymodal C-fibers

which respond to both mechanical and thermal stimuli, we observed a profound reduction of mechanosensitivity after yellow light mediated ArchT activation without any significant change in the response of the same receptor to thermal stimuli (Fig. 2). This result demonstrates that inhibition of the Schwann does not have an indirect effect on the excitability of the closely associated C-fiber ending. Furthermore, these results demonstrate how the Sox10+ Schwann cells are involved specifically in the transduction of mechanical and not thermal stimuli. Indeed these data are consistent with the well-established fact that cold and heat stimuli evoke ionic inward currents in isolated nociceptors that are probably mediated via thermo-TRPs[41–43].

We have also demonstrated that there is tight electrical coupling between Sox10+ sensory Schwann cells in Meissner's corpuscles as well as hair follicles innervated by RAMs. While blue light excitation of Sox10+ Schwann cells activated a minority of RAMs, yellow light mediated inhibition of the same cells led to a reduction in

mechanosensitivity in around half of all RAMs recorded. RAMs are equipped with KCNQ4 and $K_v1.1$ potassium channels that act as a break on excitation[44,45] which might partly explain why blue light stimulation did not always evoke a spike. The existence of RAMs not modulated by light could be due to the genetic approach in which not all Sox10[+] cells might express ChR2 or ArchT following tamoxifen induced recombination. Alternatively, there may be two populations of RAMs that are differentially connected to different subsets of cells within the corpuscle, with RAMs not directly coupled to Sox10[+] displaying no modulation. Recent work has indeed suggested that there may be two molecularly distinct RAM populations innervating Meissner's corpuscles[21]. Here we found no differences in the stimulus-response coding of RAMs that were or were not modulated by light in Sox10-ArchT or Sox10-ChR2 mice (Supplementary Fig 3c, d, g, h, Supplementary Fig. 4e, f). Each Meissner's corpuscle can be innervated by between 1 and 4 RAM axons[25], thus it is possible that each ending displays unique connectivity. Remarkably, optogenetic manipulation of Sox2[+] cells within the Meissner's revealed that these cells are coupled very differently to RAM axons compared to Sox10 cells. Thus, excitation of the Sox2 cells, only one or two of which sit at the base of the corpuscle, evoked very long latency tonic responses in some RAMs (Fig. 4). Indeed the Sox2 mode of coupling appears more analogous to that of Merkel cells to SAMs[14,15]. But even more striking was our finding that the mechanosensitvity of almost all RAMs was reduced after light induced inhibition of Sox2 cells (Fig. 4). This result in itself suggests that the genetic strategy used probably cannot account for the diversity of connectivity that we observed.

All our results strongly support the idea that a substantial part of the transduction of the mechanical stimulus into an electrical signal takes place in sensory Schwann cells. However, the nature of this transduction process may be fundamentally different between Sox10 cells associated with mechanoreceptors and nociceptors. It is clear that Piezo2 is essential for sensory mechanotransduction in many mechanoreceptors[4,6,46–49]. However, in two studies in which direct recordings were made from single mechanoreceptors it was found that many mechanoreceptors were still mechanosensitive in Piezo2 conditional knockout mice, in both cases Cre lines were used that drive recombination below cervical levels[4,46]. Using the same conditional mutant mice as Hoffman et al. recordings from the DRG made with a multi electrode array claimed absence of mechanoreceptor activity[49] and recordings from dorsal horn neurons in the same conditional mutant mice also indicated a profound loss of mechanoreceptor input[47]. Similarly, using calcium imaging methods, it was claimed an almost complete absence of mechanoreceptor function in the cervical sensory ganglia of mice in which Piezo2 was conditionally deleted using a viral approach[48]. Mechanosensitivity, is only present at the receptive field in the skin, with no direct involvement of the cell body. It seems likely that methods focused on recording from or imaging the cell bodies of sensory neurons underestimate the amount of intact transduction remaining after Piezo2 gene deletion. It is also possible that Piezo2 gene deletion or even loss of mechanosensitivity in mechanoreceptors could have indirect effects on primary afferent connectivity. Here we have not addressed the molecular nature of mechanotransduction in sensory Schwann cells. There is, however, evidence that Piezo1 may be a mechanotransducer in keratinocytes and could act to amplify mechanical nociception[17], but the mechanisms appear distinct from the nociceptive Schwann cells studied here.

In vivo Sox10[+] cells are both anatomically and functionally diverse and it is not presently possible to determine the original nature or origin of a cultured Sox10[+] cell, limiting functional characterization of specific subtypes in vitro. Furthermore, numerically speaking nociceptive Schwann cells greatly outnumber the Sox10[+] cells associated with mechanoreceptors. So far, we have little information about the molecular nature of mechanotransduction in Schwann cells associated with nociceptors or mechanoreceptors. Mice with a *HoxB8-Cre* driven

conditional deletion of *Piezo2* show a substantial loss of mechanoreceptor function and Aδ− and C-fiber nociceptors show blunted dynamic responses to noxious pressure[4]. However, there was no indication that Aδ− and C-fiber nociceptors lost their mechanosensitivity in these mice. It is possible that the Hoxb8 promotor drives recombination in Sox10+ Schwann cells which would suggest that Piezo2 in Schwann cells does not substantially contribute to nociceptor transduction, an idea that remains to be tested directly. Recent electrophysiological recordings from lamellar cells of the Grandry corpuscle, the avian equivalent of the Meissner corpuscle, revealed that these cells exhibit mechanosensitivity and are tightly coupled to the sensory ending[50,51]. The nature of the mechanosensitive channel that confer fast transduction to these cells is currently unknown. Nevertheless, the data from birds is in very good agreement with our results showing tight physiological coupling of specialized RAMs to cells within the Meissner´s corpuscle. Optogenetic inhibition of just a proportion of Meissner's corpuscles was sufficient to elevate perceptual threshold to detect a vibration stimulus. Indeed, in our goal directed task the mouse is able to detect and react to sinusoids in a time frame in which just one or two sinusoids are delivered. Sinusoidal stimuli are thought to mimic movement of the skin over rough surfaces and thus the detection of smoothness or roughness is critically dependent on Sox10[+] cells within the Meissner corpuscle.

Our results suggest that for the vast majority of sensory afferents in the skin, the properties of the sensory neuron membrane can only give a partial picture of sensory mechanotransduction. Thus, specialized glio-neural end-organs with diverse functionality appear to be integral in conferring physiological mechanosensitivity to both nociceptors and mechanoreceptors. Understanding the molecular diversity, regulation and plasticity of these functionally distinct glio-neural end-organs will be critical to learning how to treat touch and pain disorders.

## Methods

### Mouse strains

All animal work was approved by Ethical Committees on Animal Experiments. In Stockholm the Stockholm North committee and in Berlin the Landesamt für Gesundheit und Soziales (LAGeSo, State of Berlin). Mice of both sexes and from mixed background were used in this study. Animals were kept in cages in groups, with food and water *ad libitum*, under 12 h light-dark cycle conditions. Sox10[iCreERT2] mouse strains has been previously described[19]. Sox2[CreERT2] (stock number 017593), Rosa26R[tdTomato] (stock number 007914), Rosa26R[ChR2-EYFP] (stock number 012569) and Rosa26R[ArchT-EGFP] (stock number 021188) were ordered from The Jackson Laboratory. Sox10::iCre[ERT2] and Sox2[CreERT2] mice were crossed to R26R[TOM] mice for histological analysis and to R26R[ChR2] and R26R[ArchT] for Schwann cell isolation experiments and for functional experiments.

Tamoxifen (Sigma, T5648) was dissolved in corn oil (Sigma, 8267) at a concentration of 20 mg/ml and delivered by intra peritoneal (i.p.) injection to adult mice (2 consecutive injections) or pups (P10, single injection).

### Tissue preparation

Adult mice were sacrificed with isoflurane overdose and hindpaws were then collected and fixed in PFA for 24 h at 4 °C, washed 3 times with PBS and cryoprotected by incubating at 4 °C in 30% sucrose in PBS for 24 h. Plantar skin of each paw was then dissected out, embedded in OCT compound (Tissue-Tek) and frozen at −20 °C. Tissue samples were sectioned at 14 μm thickness and conserved at −20 °C until further use.

### Immunohistochemistry

Thawed sections were air dried for 1 h at room temperature (RT). Sections were then washed in PBS and incubated in blocking solution

(5% normal donkey serum (NDS, Jackson Immuno Research, #017-000-121), 2% Bovine Serum Albumin (BSA, Sigma, #A7906), 0.3% Triton X-100 in PBS) for 1 h before applying primary antibodies overnight at 4 °C. The following primary antibody (diluted in the blocking solution) was used: rabbit anti–PGP9.5 (1:400, Thermo Fisher Scientific, #PA5-29012), rabbit anti-S100β (1:500, Dako, #Z0311) and rabbit anti-Sox2 (kind gift from T. Edlund, 1:10). For sox2, sections were developed and visualized with TSA Plus kit (PerkinElmer) according to manufacturer's protocol. For detection of the primary antibodies, secondary antibodies raised in donkey and conjugated with Alexa-488 and 647 fluorophores were used (1:1000, Molecular Probes, Thermo Fisher Scientific) for 1 h at RT. DAPI staining (1 mg/ml, Thermo Fisher Scientific, #D1306) was performed as the same time as secondary antibodies. Sections were then washed 3 times with PBS and mounted using fluorescent mounting medium for imaging (Dako, #S3023). The anti-PGP9.5 antibody was verified by relative expression and has been validated in various studies[52–54]. The S100β antibody has been validated in many studies including[19,21]. The Sox2 antibody validation has been described[55].

Images were acquired using Zeiss LSM700 confocal microscope equipped with 40x objective. Images were acquired in the.lsm format and processed with ImageJ. Representative images are projections of Z-stacks taken at 1 μm intervals.

## Terminal Schwann cell dissociation and culture
Briefly, terminal Schwann cells were obtained from glabrous skin of Sox10-ChR2 P14 pups. Pups were sacrificed with isoflurane overdose; paws were quickly collected in ice cold HBSS medium (Thermo Fisher Scientific, #14170112) containing 100 U/ml penicillin, 100 ug/ml streptomycin (supplied as a mix, Thermo Fisher Scientific, #15140122). Plantar skin was then dissected out from each paw, and after removal of nerves and other tissues, skin was incubated in fresh HBSS containing 4 mg/ml of collagenase/dispase (Sigma-Aldrich, cat.11097113001) for 25 min at 37 °C. Epidermis was then removed and the dermis, after careful removal of footpads, was cut in small pieces and incubated with collagenase/elastase (Worthington. Cat. LK002066) 4 mg/ml in HBSS for 40 min at 37 °C. DNAse I was added (Worthington. Cat. LK003170) to a final concentration of 1 mg/ml before mechanical dissociation with fire polished Pasteur pipettes coated previously with 1% BSA in PBS. The cell suspension was slowly filtered through 40 μm-pore size cell strainer and centrifuged at 300 g for 6 min. The pellet was re-suspended in Schwann cell medium (DMEM) with D-valine (Miclev, #AL251) supplemented with 2 mM glutamine (Thermo Fisher Scientific, #23030081), 10% Fetal Bovine Serum (Sigma, #2442), 1% N2 (Life Technologies, #17502001), 100 U/ml penicillin, 100 μg /ml streptomycin, 5 μM forskolin (Sigma #F6886) and 20 μg/ml bovine pituitary extract (Sigma, #P1476). Cells were plated on coverslips coated with poly-L-lysine (Sigma, #P4707) for 2 h at 37 °C and then with laminin (Sigma, #L2020) for 30 min at 37 °C. Cells were cultured in humidified 5%CO₂/95% air atmosphere. Cultured cells were used for experiments between days 2 and 3 days in vitro.

## Whole-cell electrophysiology
Whole-cell patch-clamp voltage clamp recordings were performed on Sox10^ChR2 (at DIV 2–3) cultured terminal Schwann cells at room temperature (20–24 °C). Recordings from fluorescent cells were performed using Multiclamp 700B amplifier (Molecular Devices) and analyzed off-line in Clampfit software (Molecular Devices). Patch pipettes with a tip resistance of 2–3MΩ were filled with intracellular solution (in mM):105 K-gluconate, 30 KCl, 10 Na-Phosphocreatine, 10 HEPES, 4 Mg-ATP, 0.3 Na-GTP and pH adjusted to 7.3 with KOH. The extracellular solution contained (in mM): 125 NaCl, 2.5 KCl, 25 NaHCO3, 1.25 NaH2PO4, 1 MgCl2, 2CaCl2, 20 glucose and 20 HEPES. Cells were clamped to a holding potential of −40 mV and stimulated with a series of mechanical stimuli with probes (tip diameter 2–3 μm) that were

custom-made with patch pipettes heated for 10 s with a microforge (Narishige MF-90).

## Extracellular recording from tibial and saphenous nerve
Electrophysiological recordings from cutaneous sensory fibers of the tibial or saphenous nerve were made using an ex vivo skin nerve preparation following the method described previously[5,25]. Briefly, the animal was sacrificed by cervical dislocation and the hair of the limb was shaved off. The glabrous skin from the hind paw was removed along with the tibial nerve dissected up to the hip and cut. The glabrous skin along with the tibial nerve still attached to the hindpaw was transferred to a bath chamber which was constantly perfused with warm (32 °C) oxygen-saturated interstitial fluid. The remaining bones, muscle and ligament tissue were gently removed as much as possible, allowing the glabrous skin and tibial nerve preparation to last at least 6 h of recording in healthy a stable condition in an outside-out configuration. The tibial nerve was passed through a narrow channel to an adjacent recording chamber filled with mineral oil. Normally, between one and 10 neurons could be recorded per preparation, both males and female mice were used in the study. Note all single unit datasets were obtained from 13 Sox10-ChR2 mice, 13 Sox10-Cre mice, 32 Sox10-ArchT mice, 19 Sox2-ChR2 mice, and 6 Sox2-ArchT mice.

Single-unit recordings were made as previously described[5,25]. Fine forceps were used to remove the perineurium and fine nerve bundles were teased and placed on a platinum wire recording electrode. Mechanical sensitive units were first located using blunt stimuli applied with a glass rod. The spike pattern and the sensitivity to stimulus velocity were used to classify the unit as previously described[5,25]. Raw data were recorded using an analog output from a Neurolog amplifier, filtered and digitized using a Powerlab 4/30 system and Labchart 8 software with the spike-histogram extension (ADInstruments Ltd., Dunedin, New Zealand). All mechanical responses analyzed were corrected for the latency delay between the electrical stimulus and the arrival of the action potential at the electrode. The conduction velocity (CV) was measuring the formula CV = distance/time delay, in which CVs >10 ms⁻¹ were classified as RAMs or SAMs (Aβ, <10 ms⁻¹ as Aδ and <1 ms⁻¹ as C-fibers).

## Mechanical stimulation
Mechanical stimulation of the receptive field of the recorded fibers was performed using a piezo actuator (Physik Instrumente, Germany, P-602.508) connected to a force measurement device (Kleindiek Nanotechnik, Reutlingen, Germany, PL-FMS-LS). Different mechanical stimulation protocols were used to identify and characterize the sensory afferents. Mechanoreceptors were tested with a vibrating stimulus with increasing amplitude and 20 Hz frequency. The force needed to evoke the first action potential was measured. Additionally, a ramp and hold step was used with Constant force (100mN) and repeated with varying probe movement velocity (0.075, 0.15, 0.45, 1.5 and 15 mm s⁻¹). Only the firing activity evoked during the dynamic phase were analyzed. SAM mechanoreceptors and nociceptors were tested with a mechanical stimulus with a constant ramp (1.5–2 mN ms⁻¹) and increasing force amplitude, spikes evoked during the static phase were analyzed.

## Thermal stimulation
Thermal stimulation was carried out in two ways. First, a qualitative classification of C-fiber nociceptors was made applying cold and hot SIF buffer directly to the receptive field of the terminal ending which was isolated by metal ring. Cold buffer was kept on ice at 4 °C and reach -10 °C at stimulation. Hot buffer was kept in a shaker incubator at 80 °C and skin temperature reached ~50 °C during stimulation. Thereafter, a custom designed thermostimulator connected to a thermocouple and Peltier that could be placed in direct contact with the skin was used. Two sequential temperature ramps were applied to test the thermoreceptors sensitivity. First, a cold ramp starting at 32 °C

(the skin basal temperature) and decreasing in 2 degrees per second rate until reaching 12 °C and coming back to 32 °C as fast as possible. Thereafter, a heat ramp was applied starting from 32 °C with an increasing temperature of 2 degrees per second until reaching 52 °C and coming back to 32 °C as fast as possible. A gap of 30 s between the two thermal ramp stimulation was used for sensory afferents to recover.

The piezo actuator, thermostimulator, and optogenetic lamp for blue and yellow light were connected to a micromanipulator for positioning.

### Excitatory optogenetic

Cultured Schwann cells or receptive fields in the skin-nerve preparation were stimulated with blue light (470 nm, 5 s on for patch-clamp or 10 s on for skin-nerve preparation) applied through a flexible optical fiber bundle perpendicular to the skin. Light was applied with increasing intensity of 0.5, 2.6, 3.9 and 4.3 mW/mm$^2$ once the responsive single-unit was isolated.

### Inhibitory optogenetics

Sensory neuron terminals were identified by mechanical stimulation of the their receptive field using a glass rod and subsequently classified as mechanoreceptors or nociceptors accordingly to their responses to standardized stimuli and their conduction velocity[25]. Yellow light was applied (575 nm, 5 s ON 1 s OFF) for 10 min after determination of the baseline mechanosensitivity of the receptor. A light intensity of 0.5 mW/mm$^2$ was used for the inhibition protocol. Vibration stimuli or ramp and hold stimuli were used to evaluate mechanosensitivity of the same unit before and after light exposure. Mechanosensitivity was evaluated 5 min for mechanoreceptors or every 10 min for nociceptors for 15 or 20 min, respectively. All sensory afferents were characterized according to their responses to increasing velocity, vibration and/or ramp-and-hold mechanical stimuli to evaluate their adaptation properties.

### Surgery for behavioral training

Mice were anesthetized with isoflurane (3–4% initiation, 1.5–2% maintenance in O2) and had a subcutaneous injection of Metamizol (200 mg per kg of body weight). The temperature of the animals was monitored at all times and kept at 37oC using a heating pad. A light metal support was implanted onto the skull with glue (UHU dent) and dental cement (Paladur). Mice recovered in their home cage and had Metamizol in the drinking water (200 mg/mL) for 1–3 days. Both male and female mice were used (6 males, 3 females).

### Go/No Go vibrotactile detection task

Head implanted mice underwent habituation to the behavioral setup at several times for 4 days with gradually increasing head restraint periods (5–60 min). The right forepaw was tethered with medical tape to a glass surface with a small hole, through which the vibrotactile stimulator could contact the center of the forepaw. The vibrotactile stimulator (smooth plastic cylinder ~2 mm diameter, driven by Dual-Mode Lever System 300 C, Aurora Scientific) therefore touched the right forepaw glabrous skin. During habituation, the animals were occasionally rewarded with condensed milk droplets to reduce stress.

Next, the animals were water restricted and underwent two pairing sessions (30–60 min) in which vibrotactile stimuli were presented simultaneously with water rewards (4–7 µl) coming from a lick spout, in order to form an association between stimulus and reward. After the pairing, mice were trained on consecutive days to lick the water spout in response to a vibrotactile stimuli (1 s long, increases in force of up to 15–20 mN at 20 Hz, starting from a constant baseline of 9 mN), with water rewards being given when the mice licked during the stimulus presentation. Performance was assessed by comparing the correctly reported stimulations (hits) with spontaneous licking occurring during

catch trials (no-stimulus time windows of the same length as stimulus trials) (false alarms).

Mice that reached a d' > 1.5 (see analysis of behavior) were then trained with shorter stimuli of lower amplitudes (0.4 s long, interleaved vibrotactile stimuli of 20 Hz and 1.5 or 3 mN amplitude from a 9 mN baseline) until they reached d' > 1.5 in these conditions. Each training session on low amplitudes lasted for 200 trials, with equal proportions of each amplitude and catch trials.

To inhibit Sox10-ArchT cells before behavioral testing, we temporarily replaced the vibrotactile stimulator with a yellow LED light (Thorlabs). The LED light illuminated the paw from a distance of ~3 cm below the glabrous skin. The equipment and light parameters were the same as in ex-vivo recordings, with the only difference being the duration of light stimulation (30 min). The increased stimulation time in behavior aimed to suppress Sox10+ cell activity for a longer time, since a behavioral session lasted for ~40–60 min. The paw and LED were covered with optical blackout cloth (Thorlabs) in order to prevent the mice from seeing the light, and illuminating any other body region of the mice. To assess whether the observed effect was dependent on the stimulated skin region, we stimulated the untrained, contralateral (left) forepaw and saw no effect.

### Analysis of behavior

Licks of the water spout were measured using a capacitance sensor. The performance during the detection task was assessed by comparing the hit (% of reported stimulus trials) with false alarm (% of reported catch trials) rates. Each trial consisted of a pre-stimulus window of 0.5 s, followed by a stimulus (or catch) window of 0.4 s. Licks after this time window were excluded for performance analysis. Trials were delayed by a random interval between 3 and 30 s if mice spontaneously licked during a 2 s window before the start of a new trial. All mice had to detect interleaved stimuli of 1.5 and 3 mN. Performance data for each mouse was obtained for trials of 1.5 and 3 mN before and after the optogenetic manipulation. Since some mice showed a deficit when reporting both 1.5 and 3 mN while others only showed an effect for 1.5mN, the analysis was carried out on the largest amplitude affected in each mouse. 1/7 mice did not meet the perceptual performance criteria for optogenetic testing, and one further mouse was excluded from analysis because it did not show any effect of optogenetic manipulation most likely because of variability of ARCH expression during tamoxifen induction.

To compare performance between different mice and training sessions, we used d' (sensitivity index) instead of the % of correct trials, in order to account for bias in the licking criterion. Sensitivity was calculated with the formula d' = z(h) – z(fa), where z(h) and z(fa) are the normal inverse of the cumulative distribution function of the hit and false alarm rates, respectively.

The z scores for hit and false alarm rates were calculated with OpenOffice Calc (Apache Software Foundation) using the function NORMINV.

The behavioral training was run with Bpod (Sanworks) and data was collected with custom-written routines in MATLAB (Mathworks). Custom-written MATLAB and Python (Python Software Foundation) scripts were used for analysis.

### Statistical tests

Statistical analyses were carried out with GraphPad Prism 5.0/6.0 and Python. Statistical tests for significance are stated in the text, and include Mann–Whitney test, Wilcoxon matched pairs test and Student t-test and two-way ANOVA. Asterisks in figures indicate statistical significance: *$P < 0.05$, **$P < 0.01$, ***$P < 0.001$.

### Quantification and statistics

In the patch clamp experiments the tau (τ) value of activation and inactivation of a current trace was calculated as exponential fit of

different phases using Clampfit 10.7 software. To calculate *P*-value and statistical significance unpaired *t*-test was performed.

In the skin nerve experiments, raw data were stored and processed using Microsoft Excel. Statistical tests were performed using Prism 8 (GraphPad Software, San Diego, CA, USA). Data were tested for normality. Light-responsive and non-responsive sensory afferents, and light- and mechanical response from the same sensory afferent, were compared using unpaired Student's *t*-tests or multiple comparisons two-way repeated-measures analysis of variance (ANOVA) and post-hoc tests performed with Bonferroni's multiple comparisons test. Significance values are reported as: $*P$-value $\leq 0.05$; $**P$-value $\leq 0.005$; $***P$-value $\leq 0.0005$. All error bars are standard error of the mean (SEM).

### Reporting summary
Further information on research design is available in the Nature Portfolio Reporting Summary linked to this article.

### Data availability
All data underlying the manuscript will be made available on request. Source data is provided. Source data are provided with this paper.

### Code availability
The custom MATLAB and Python scripts will be shared by the authors upon request.

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

## Acknowledgements

This research was funded by the ERC (DescendPain 101053091 to P.E., Sensational Tethers 789128 and Deutsche Forschungsgemeinschaft CRC 958 grant to GRL and ACoolTouch 682422 to JFAP) the Swedish MRC (2019-00761 to P.E.), KAW Scholar and project grant, Wellcome Trust (200183) to P.E. LCE was supported by a Ramon y Cajal grant (RYC2021-034520-I). R.K. was supported by a DBT Ramalingaswami Fellowship (BT/HRD/35/02/2006), IBRO Return Home Fellowship and ISN-CAEN Return Home Grant. MDZ was supported by a Brain Foundation and Swedish Society for Medical Research (SSMF) post-doc fellowship. We thank members of the Lewin and Ernfors lab for constructive comments on the manuscript.

## Author contributions

Conceptualization: G.R.L. and P.E.; mouse models/experimental design: J.O-A. and L.C.-E.; nerve recording and formal analysis: J.O.-A.; mouse behavior R.P.-M. and J.F.A.P.; immunohistochemistry: L.C.-E. and M.-D.Z.; patch clamp electrophysiology: R.K.; writing (J.O.-A., L.C.-E., P.E., and G.R.L.) with input from all authors; and supervision and funding: G.R.L. and P.E.

## Funding

## Competing interests

The authors declare no competing interests.
