## [Peer Review File · Nature Communications]

Sensory Schwann cells set perceptual thresholds for touch and selectively regulate mechanical nociceptionREVIEWER COMMENTS

Reviewer #1 (Remarks to the Author):

- This is a follow-up study to a 2019 publication (in *Science*), which had described that specialized cutaneous Schwann cells (that express Sox10) form direct excitatory connections with sensory neurons and that these Schwann cells are mechanosensitive and transmit nociceptive information to sensory nerves. The present study is an extension of these findings, where the authors state that Schwann cells associated with Meissner corpuscles are necessary for normal vibrotactile stimulus-induced firing in RAMs. The authors find that optogenetic inhibition of Sox10+ Schwann cells leads to impairments in a goal-oriented tactile perception task using vibro-tactile stimuli. The authors also describe that Schwann cells are necessary for mechanical sensation, but not thermal sensation, in several types of nociceptor neurons using an *ex vivo* electrophysiology preparation.
- An advance of this study is that Sox10+ Schwann cells are required for normal RAM responses to vibrotactile stimuli during a behavioral assay. The authors had previously found that Schwann cell activation increases behavioral responses to mechanical, nociceptive stimuli (2019 study). In this study, the authors perform *ex vivo* electrophysiology recordings to confirm this prior finding, and also demonstrate here that Schwann cell activation does not impact thermal sensitivity of C-fiber nociceptors.
- Overall, the findings of this study are interesting and contribute to our understanding of neuronal-glia interactions for transmission of tactile and nociceptive stimuli from the skin. There are some questions regarding experimental findings and interpretations which are described below.
- Some of the interpretations are slightly overstated and could be modified. For example, sensory neuron responses to mechanical stimuli versus optogenetic activation of Schwann cells are clearly different (which is interesting). While optogenetic inhibition of Schwann cells alters the firing rates of sensory neurons, the sensory neurons still fire and exhibit some response to mechanical stimulation (e.g., Figures 2, 3). Thus, it would be more appropriate to conclude that sensory Schwann cells in the Meissner's corpuscles are required for normal vibration sensing. In addition, Lines 300-302: "These results necessitate a major realignment of somatosensory research away from the primary afferent neuron towards specialized glio-neural end-organs in the skin." This statement is a bit extreme. It seems that somatosensory research should not be directed away from primary afferent neurons, but rather, should also include investigations of specialized glio-neural end-organs in the skin. Direct manipulations of sensory neurons will also cause substantial changes in tactile sensitivity and behavior.
- There are some questions surrounding optogenetic manipulation of Schwann cells and the variability/efficacy of this approach. In Figure 2E, the latency in A-M neuron responses to light are highly variable (~0 second – over 1 second). This sort of variability is also observed in C-fibers in the same panel. However, the response latency for each neuron type to mechanical stimulation is not nearly as variable. What accounts for the high variability in response latency to light in these neurons? Similarly, it is unclear why only a minority of RAMs are modulated by optogenetic inhibition or activation of SOX10+ Schwann cells. The authors state that "In most Meissner's corpuscles we found 2-4 Sox10-TOM+ cells to be intimately associated with the sensory endings of rapidly-adapting mechanoreceptors (RAMs)". If most Meissner corpuscles contain 2-4 Sox10-TOM+ cells that associate with RAMs, why don't the optogenetic manipulations impact most RAM neurons? The authors mention that "probably no more than 80% of the Sox10+ cells would express ChR2 or ArchT." Quantifications of Sox10+ cells that express ChR2 or ArchT would be helpful for understanding this discrepancy. It would also be helpful to show a larger region of skin to demonstrate specificity of Sox10 expression with Schwann cells (and that there is no expression in other cell types in the skin region of interest).
- Does optogenetic activation or inhibition alter the threshold for spiking in either nociceptor neurons or RAMs? This information could be gleaned from the existing data, and would be helpful for understanding the mechanisms through which Schwann cells contribute to mechanical nociception and tactile sensations.

- It is unclear why 10 minutes (or more) of optogenetic inhibition is required for the experiments in Figures 2, 3 and 4. Control experiments are important to ensure that ArchT is having the desired impact on this precise cell type only (i.e., optogenetic inhibition of Schwann cells should impair the generation of transient currents by mechanical stimulation), as were performed with optogenetic activation of Schwann cells in Figure 1.
- It would be very helpful if the data showing a lack of SAM response to optogenetic inhibition of SOX10+ Schwann cells were included (Lines 223-226). This would support specificity of the mouse genetic strategy.
- Figure 4F: it appears that optogenetic inhibition of Schwann cells impairs vibrotactile stimulus detection but this effect does not persist for all of the trials. The animal seems to recover normal performance by trial ~40. Is the optogenetic effect transient? This raster plot does not seem to match the quantified data in Figure 4D? Why are only a subset of trials show for Figure 4F, compared to those analyzed in Figure 4D?
- Figure 4H: the authors performed paired t-tests for analysis of this dataset. A repeated measures, two-way ANOVA with post-hoc test would be more appropriate for this experiment (or at least some other analysis method that considers multiple comparisons). Is there no statistically significant difference in d' between 'after' and 'recovery' phases? If not, this needs to be stated and explained.

Reviewer #2 (Remarks to the Author):

The paper follows a previous publication (Abdo et al., Science 365, 695–699, 2019) from the same group that identified a novel peripheral glial cell subtype. The previous paper proposed the existence of specialized Schwann cells on the very terminal region of cutaneous primary sensory neurons that, with their extensive processes, signal thermal and mechanical noxious stimuli to the neurons. The paper further investigates with more detailed findings the different types of sensations that nociceptive Schwann cells convey to specific subsets of primary sensory neurons. A major observation is that nociceptive Schwann cells also surround the nerve-endings forming Meissner's corpuscle, thus being implicated in the perception of vibrotactile stimuli.

It would be helpful for readers if the authors better clarified that the original observation reported in the previous paper (Abdo et al., Science 365, 695–699, 2019), that nociceptive Schwann cells mediate both mechanical and thermal stimuli, is only partially confirmed in the present study, which shows that only mechanotransduction implicates a role of nociceptive Schwann cells.

The authors should discuss the finding that half of C-M and A-M are non-responsive to blue-light. Are these non-responsive nerve fibers not surrounded by nociceptive Schwann cells? This question may also apply to the few non-responsive polymodal C-fibers, most RAM and all SAM. The findings seem to imply that two populations of DRG neurons exist, one with nociceptive Schwann cells on their terminal fibers and another without. If this is the case, what are the differences between the two populations in signaling innocuous and noxious stimuli?

The number of mice varies between 4 and 5. Are these numbers sufficient? And how was the sample size calculated as the numerosity varies across experiments?

Line 86: a brief description of the method would help the readers.

The authors usually use $P <$, but sometimes (line 65, 199) they use $P =$. What is the reason for the difference?

Sentences like 'Schwann cells tended to show larger currents in response 66 to mechanical compared to blue light stimulation, but this not statistically different.' should be avoided.

Minor:

SAM should be spelled out at its first appearance (line 163).

Refs 5 and 7 are the same.

There are typos and unclear sentences. For example:

Line 653: a profound decreases

Line 213: We measured the force amplitude for the first spike as well as frequency following (where 1.0 denotes a spike evoked by every sinusoid) before and ten minutes after cyclical yellow light was focused on the receptive field

And others throughout the paper.

Reviewer #3 (Remarks to the Author):

The detection of mechanical forces relies on the activation of sensory neurons that innervate the body and connect with the central nervous system. Somatosensory neurons are functionally and molecularly diverse with individual gene expression profiles tightly correlated with receptive tuning properties. The consensus view is that signal transduction occurs at the sensory neuron terminal. Consistent with this idea is the finding that sensory neurons express receptors that gated by environmental stimuli and these molecules are required for touch and temperature sensation.

It is also well established that sensory neuron endings are embedded within tissue where they form highly specialized endings and interact with other cells, including epithelial and glial cells. For example, in skin, Merkel cells are mechanosensory epithelial cells that contribute to touch responses. There is also evidence that keratinocytes play a role in mechanosensation. More recently, work from the Ernfors group has found that specialize glial cells called Terminal Schwann cells are required for normal sensory responses. This latter work led to the Ernfors group to postulate that terminal Schwann cells are bonafide sensory cells, challenging the idea of the passive/supportive roles typically assigned these cells.

While the notion that Schwann cells are principle detectors of mechanical stimuli is a very interesting hypothesis, it has remained unclear whether they have 'instructive' vs 'permissive' roles. Here, Ojeda-Alonso and colleagues have set-out to provide a causal link between terminal Schwann cell stimulation and sensory detection. They provide evidence that terminal Schwann cells are intrinsically mechanosensitive and that their direct stimulation is necessary and sufficient to drive sensory neuron firing in an ex vivo preparation. Lastly, they explore whether optically perturbing Schwann cell function alters perceptual thresholds.

Overall assessment:

This is undoubtedly an intriguing and potentially important area. However, there are several key experiments missing that are needed to substantiate this study's conclusions. Specifically, better characterization of the mechanically evoked currents, validation of the inhibitory opsins effects on the Schwann cells and a positive control for the behavioral experiments are required. Lastly, the study presents an unbalanced view. The manuscript neglects to mention many well-established findings including the contributions of non-neuronal cells (for example, the work of Stucky, Lumpkin and Caterina Labs) and the fact that removing Piezo2 from sensory neurons completely blocks many aspects of touch perception (for example the work from Patapoutian, Chesler and Ginty labs). Furthermore, the references selectively neglect some groups while favoring others. The authors should provide a more scholarly and balanced view of the field. Last, the conclusions are out of proportion to the data. The effect sizes are relatively small in comparison to those seen with sensory ion channel knockouts or neuron ablation experiments. More measured and sober conclusions are warranted.

Specific comments:

In Figure 1, the authors record from a small number of cells and only provide coarse information about the currents. The trace is low resolution. Shorter indentations (50-250msec) done in a stepwise series is need to better understand the kinetics and sensitivity of the evoked current. Given this is a 'new' type of mechano-current, the authors should show what is the voltage dependence, reversal, and ion permeability. Presumably this is a cation channel and excitatory?

The authors should do current clamp recordings. Are Schwann cells excitable? The authors should demonstrate if they fire APs when stimulated. It seems like their resting potential is around -30 mV and their mechano-currents are small, so it is hard to understand how they might signal to an adjacent nerve ending. Similarly, it would be nice to see how ChR2 affects the membrane potential of these cells – are these cells excitable (possibly firing calcium spikes?) or would we just see a depolarization.

Much of the study relies on Arch for inhibition yet no characterization of this tool in Schwann cells is provided. It is unclear if/how Arch inhibits Schwann cells. The authors should perform some *in vitro* characterization of the effects of Arch on Schwann cell physiology. For example, it would be nice to see current clamp experiments showing if Arch hyperpolarizes the membrane and how long this takes. It is unclear how activation or inhibition of Schwann cells modulates the coupling between Schwann cells and nerve terminals. While dissecting out the nature of this coupling is well beyond the purview of this paper, some insight into the effects of the two opsins on Schwann cell membrane potential would help us understand how these manipulations affect the skin nerve prep recording or behavioural assays.

The optical excitation of Schwann cells in the skin-nerve recordings is very nice. However, it would be better if the authors could provide a better idea of how tightly coupled things are. Does the spiking in the nerve correlate with the intensity, duration, and frequency of the optical stimulation of the Schwann cells? How repeatable is this (e.g. what if you stimulated 5 times in a row?). Does it fatigue during longer stimulation? Minor note- presumably the authors have made sure there is not any ectopic expression of ChR2 in sensory neurons due to low levels of Sox10 expression during development.

In Figure 2I-L and similar, the summary data are highly processed. We would have more confidence in the authors' conclusions if we could see the absolute numbers for spikes and threshold before and after illumination in the control and Arch mice. This should be represented with the individual replicates. This would allow the reader to better gauge the variability across units and assess for themselves the size and importance of the effects of Arch inhibition on stimulus evoked firing. The authors should explain in the main body of the text the criteria and procedure for including and excluding units in Figures 2J-L – it seems like the authors only quantified the units where Arch had an inhibitory effect, however the reader needs confidence this was done in an objective manner, to avoid biasing results. How did Arch affect the units that did not reduce their firing – did these units show no effect, or did they perhaps increase their firing – extended data showing these units would be very helpful.

The representative image in Fig 3 should be improved to more convincingly show this is a Meissner corpuscle and to better delineate the nerve endings and the Sox10. At least in the downloaded version the green is impossible to see, and the red is diffuse. The DAPI image does not add much.

The behavioral data show some effect, but it is clear the mice can still detect the stimuli. The conclusions therefore need to be toned down quite a bit. Arch-negative mice should be included to control for the long light stimulation/paw restraint.

REVIEWER COMMENTS

Reviewer #1 (Remarks to the Author):

This is a follow-up study to a 2019 publication (in *Science*), which had described that specialized cutaneous Schwann cells (that express Sox10) form direct excitatory connections with sensory neurons and that these Schwann cells are mechanosensitive and transmit nociceptive information to sensory nerves. The present study is an extension of these findings, where the authors state that Schwann cells associated with Meissner corpuscles are necessary for normal vibrotactile stimulus-induced firing in RAMs. The authors find that optogenetic inhibition of Sox10+ Schwann cells leads to impairments in a goal-oriented tactile perception task using vibro-tactile stimuli. The authors also describe that Schwann cells are necessary for mechanical sensation, but not thermal sensation, in several types of nociceptor neurons using an *ex vivo* electrophysiology preparation.

An advance of this study is that Sox10+ Schwann cells are required for normal RAM responses to vibrotactile stimuli during a behavioral assay. The authors had previously found that Schwann cell activation increases behavioral responses to mechanical, nociceptive stimuli (2019 study). In this study, the authors perform *ex vivo* electrophysiology recordings to confirm this prior finding, and also demonstrate here that Schwann cell activation does not impact thermal sensitivity of C-fiber nociceptors.

Overall, the findings of this study are interesting and contribute to our understanding of neuronal-glia interactions for transmission of tactile and nociceptive stimuli from the skin. There are some questions regarding experimental findings and interpretations which are described below.

We appreciate that the reviewer has read our paper carefully and confirms that the results are new and interesting.

Some of the interpretations are slightly overstated and could be modified. For example, sensory neuron responses to mechanical stimuli versus optogenetic activation of Schwann cells are clearly different (which is interesting). While optogenetic inhibition of Schwann cells alters the firing rates of sensory neurons, the sensory neurons still fire and exhibit some response to mechanical stimulation (e.g., Figures 2, 3). Thus, it would be more appropriate to conclude that sensory Schwann cells in the Meissner's corpuscles are required for normal vibration sensing.

We agree with the reviewer that our data should be interpreted more carefully. We believe like the reviewer that the sensory Schwann cells contribute substantially to the mechanosensitivity, and especially the sensitivity of the glo-sensory ending. However, the sensory ending itself is still mechanosensitive after inhibition of the sensory Schwann cell. We have therefore changed the title to “Sensory Schwann cells set perceptual thresholds for touch and selectively regulate mechanical nociception” We have also toned down our interpretation in much more extensive discussion of the data in the revised manuscript.

In addition, Lines 300-302: “These results necessitate a major realignment of somatosensory research away from the primary afferent neuron towards specialized glo-neural end-organs in the skin.” This statement is a bit extreme. It seems that somatosensory research should not be directed away from primary afferent neurons, but rather, should also include investigations of specialized glo-neural end-organs in the skin. Direct manipulations of sensory neurons will also cause substantial changes in tactile sensitivity and behavior.

We agree with the reviewer and have replaced the sentence with the following “ Our results suggests that for the vast majority of sensory afferents in the skin, the properties of the sensory neuron membrane can only give a partial picture of sensory mechanotransduction.. Thus, specialized glo-neural end-organs with diverse functionality appear to be integral in conferring physiological mechanosensitivity to both nociceptors and mechanoreceptors”

2) There are some questions surrounding optogenetic manipulation of Schwann cells and the variability/efficacy of this approach. In Figure 2E, the latency in A-M neuron responses to light are highly variable (~0 second – over 1 second). This sort of variability is also observed in C-fibers in the same panel. However, the response latency for each neuron type to mechanical stimulation is not nearly as variable. What accounts for the high variability in response latency to light in these neurons?

The reviewer is right that the response latencies between optogenetic stimulation and mechanical stimulation are different (much more variable with light, Fig 2E, New Figure 1E). However, it should be noted that the data is plotted on a logarithmic scale and if one looks carefully at the latency data it can be seen that for 7 out of the 12 neurons latencies following optogenetic activation were well under 10 msec, much faster than the mechanical evoked spikes. The latency for mechanically evoked spikes is of course

determined primarily by the mechanical threshold for activation as it is only after the mechanical threshold is exceeded that spikes can be initiated. The rate at which the force increases is thus the main determinant of the latency. In contrast, light presumably depolarized the Schwann cell membrane and in the majority of cases activates the axons within a few milliseconds which means in the majority of cases there might be a direct electrical coupling between the Schwann cell and the sensory neuron membrane. In a minority of cases longer latency spikes were seen and this could be caused by a slower light-induced depolarization of Schwann cells due to a deeper location or other biological or technical variability in light intensity reaching the Schwann cell. For example, it is known that A-M neurons branch multiple times to form a large receptive field in the skin (Light and Perl 1981). Thus, it is necessarily the case that each A-M neuron has multiple branched endings that are associated with multiple sensory Schwann cells. Therefore, in some cases action potentials could be evoked almost simultaneously in two branches, these action potentials could thus collide with each other at a proximal branch point thus eliminating the fastest latency spikes. In essence if one looks at the distribution of latencies for C-fibers the picture is very similar to that seen for A-Ms with the majority of C-fibers also showing very fast responses to light.

Similarly, it is unclear why only a minority of RAMs are modulated by optogenetic inhibition or activation of SOX10+ Schwann cells. The authors state that “In most Meissner’s corpuscles we found 2-4 Sox10-TOM+ cells to be intimately associated with the sensory endings of rapidly-adapting mechanoreceptors (RAMs)”. If most Meissner corpuscles contain 2-4 Sox10-TOM+ cells that associate with RAMs, why don’t the optogenetic manipulations impact most RAM neurons? The authors mention that “probably no more than 80% of the Sox10+ cells would express ChR2 or ArchT.” Quantifications of Sox10+ cells that express ChR2 or ArchT would be helpful for understanding this discrepancy. It would also be helpful to show a larger region of skin to demonstrate specificity of Sox10 expression with Schwann cells (and that there is no expression in other cell types in the skin region of interest).

We have now included overview pictures of Meissner corpuscle receptor endings that are positive for the Sox10-tdTomato reporter. As can be seen in revised Figure 3 most of the Meissner’s corpuscles that were identified by S100 β staining were also tdTomato positive. However, it was impossible to tell in these experiments if all the lamellar cells within the corpuscle were positive or not. In some cases, S100 β positive cells within the corpuscle were only weakly positive for tdTomato. Thus, it is entirely possible that the genetic

strategy used here leads to a somewhat heterogenous expression level of the ChR2 protein in the Sox10-positive cells. This may lead to excitation of a limited number of sensory Schwann cells that are insufficient to drive enough depolarization for the afferent ending to initiate a spike. However, it should be noted that failure to see a spike with blue light stimulation does not show that there is no connection, merely that the connection is insufficient to initiate a spike. Indeed we know that RAMs especially are equipped with KCNQ4 and perhaps Kv1.1 channels that can put a strong break on spiking (see revised Discussion). Additionally, we consistently saw a larger connectivity using yellow light that inhibited the mechanosensitivity of around half of all RAMs (see revised Discussion). For the electrophysiology experiments tamoxifen injections were carried out in young adult mice and we did note here that the degree of recombination was lower than that seen when we evaluated Sox10-tdTomato reporter where tamoxifen was injected in neonatal mice (P14). We used later tamoxifen injections for the electrophysiological and behavioral experiments for logistical reasons as we had to wait several weeks before the electrophysiological or behavioral experiments could be started.

In the revised manuscript we have now added new data concerning a sub-population of sensory Schwann cells that were positive for Sox2. Using a Sox2:cre mouse we noted that Sox2 positive cells were found within the Meissner corpuscle, but there was mostly just one single Sox2 cell in the corpuscle located at the base. Note that Sox2 cells were specific for mechanoreceptors (Merkel cells and Meissner's corpuscles and were not found to be associated with nociceptor endings). We hypothesized that these cells may have a different connectivity with the endings of Meissner mechanoreceptors. We examined this issue using the same optogenetic approach as we took for Sox10 cells. This data has all been included in two new figures (Figure 4 and 5). First, we saw that a small proportion of Meissner's corpuscles in Sox2-ChR2 mice could be excited by blue light, but in a manner completely different from that seen in Sox10-ChR2 mice. In around 25% of the cells blue light drove long latency tonic firing in these fibers (revised Figure 5). Further data using Sox2-ArchT mice showed that these cells also when inhibited reduced the sensitivity of almost all RAMs (revised Figure 5). This new data suggests that there may indeed be two functionally distinct RAMs that innervate the Meissner's corpuscle that have distinct connectivities.

3) Does optogenetic activation or inhibition alter the threshold for spiking in either nociceptor neurons or RAMs? This information could be gleaned from the existing data,

and would be helpful for understanding the mechanisms through which Schwann cells contribute to mechanical nociception and tactile sensations.

We measured the mechanical threshold for activation for both mechanoreceptors and nociceptors after optogenetic inhibition and the data is plotted in revised Figures 1, 2, 3 and 5. In the case of nociceptors we observed an apparent large increase in threshold for AMs immediately after yellow light, but this was not statistically different from controls. In the case of C-fibers there was clearly no change in mechanical threshold, but spiking rates were significantly reduced (Figure 1). In contrast, for mechanoreceptors, there was clear increases in the mechanical threshold after optogenetic inhibition and indeed this appeared to be the main effect of Schwann cell inhibition. This was also the case when we used yellow light to inhibit Sox2 cells.

In the case of optogenetic excitation we did measure the mechanical threshold of fibers before and after blue light stimulation and saw no consistent change. We only have data from three units which is shown in extended data fig 4 E,F

4) It is unclear why 10 minutes (or more) of optogenetic inhibition is required for the experiments in Figures 2, 3 and 4. Control experiments are important to ensure that ArchT is having the desired impact on this precise cell type only (i.e., optogenetic inhibition of Schwann cells should impair the generation of transient currents by mechanical stimulation), as were performed with optogenetic activation of Schwann cells in Figure 1.

In order to obtain statistically robust results, we decided to use the protocol indicated. We did not know a priori how long it would take for the activation of the ArchT proton pump to inhibit the sensory Schwann cell sufficiently to reduce the sensitivity of the primary afferent. Previous behavioral experiments described by Abdo et al., however indicated that 30 minutes of yellow light was sufficient to observe effects on nociceptive behavior. We wanted to make a clean statistical comparison between yellow light used in mice where the ArchT was expressed in sensory Schwann cells and where it was not (the control situation). It is in the nature of such an experiment that sensory responses must be compared before and after optical stimulation and to do this we must mechanically stimulate the receptor repeatedly. However, repeated stimulation can sometimes lead to a desensitization of the receptor that could in principle be mistaken for a “real” optogenetic inhibition of the receptor. Therefore, in our very extensive study we systematically kept to our standard protocol that allowed us to make robust statistical

comparisons between the experimental and control situation. In the two published studies in which optogenetic inhibition of skin cells was used the study design was very different and optical inhibition of control mechanoreceptors lacking ArchT expression was not compared (e.g. in PMID24717432). Instead, for example in the paper from Maksimovic just three mechanoreceptors were examined and extremely long mechanical stimuli were applied (3min). In these three mechanoreceptors firing rates were simply compared for the same neuron during epochs of yellow light or no light (PMID: 24717432). Indeed, statistical differences were only observed when they excluded two outlier observations. The authors in that study also do not explicitly mention if they encountered mechanoreceptors that were not inhibited by yellow light. In another study where the role of keratinocytes was examined a different optogenetic strategy was used (the light activated chloride pump NpHR was expressed in keratinocytes and in Merkel cells PMID 26329459). The response of slowly adapting mechanoreceptors was assessed in the same neurons in the absence of light and then shortly afterwards with the same mechanical stimulus in the presence of yellow light (laser applied 1 s before the mechanical stimulus started). In the example traces the results seem very convincing, but in this study no statistical test was made to ascertain that the reduction in the presence was statistically different from giving two stimuli to the mechanoreceptor in animals where the NpHR was not present. The reviewer is right that a simultaneous application of light during the mechanical stimuli would have been desirable, but this was difficult to physically implement in a way where we could be sure that enough light was delivered to the Sensory Schwann cell. Furthermore, we designed the experiment so that we could make robust statistical comparisons to get an accurate estimate the real size of the effect.

In the revised manuscript we have removed the data showing mechanosensitive currents in isolated sensory Schwann cells. There are two main reasons for this. Our in vivo data clearly show that sensory Schwann cells associated with mechanoreceptor are morphologically and functionally different from those associated with nociceptor endings. These nociceptor Schwann cells massively outnumber those associated with mechanoreceptors so that the cultured Sox10 positive cells from the skin would be expected to be almost all nociceptive Schwann cells and very rarely if ever cells that were in vivo associated with mechanoreceptors. The second reason was that we tried to repeat these experiments, but the Sox10 positive cells in culture are extremely small and flat and do not look healthy. It was therefore very difficult to obtain convincing physiological data

from these cells. Although we are convinced that these cells are mechanosensitive we believe that the data is not convincing enough for us to show it in this publication.

5) It would be very helpful if the data showing a lack of SAM response to optogenetic inhibition of SOX10+ Schwann cells were included (Lines 223-226). This would support specificity of the mouse genetic strategy.

We have included this data in the new manuscript (Extended Data Fig. 6). This was a very consistent results and we have now even added new data on Sox2 positive cells (which includes Merkel cells). Thus, stimulation of the skin of mice where Sox cells express channelrhodopsin produces a tonic discharge in SAM mechanoreceptors (Sox2-ChR2 mice). This serves as a positive control to compare with results from Sox10-ChR2 mice.

6) Figure 4F: it appears that optogenetic inhibition of Schwann cells impairs vibrotactile stimulus detection but this effect does not persist for all of the trials. The animal seems to recover normal performance by trial ~40. Is the optogenetic effect transient? This raster plot does not seem to match the quantified data in Figure 4D? Why are only a subset of trials show for Figure 4F, compared to those analyzed in Figure 4D?

The reviewer makes an interesting point and is correct that the performance of the mice seems to improve with trial number. In Fig 6D (right panel), for example, the hit rate increases over trials while the false alarms stays the same. It is difficult to say exactly why this is the case, it could indeed be because the optogenetic effect is wearing off or perhaps because mice are relearning the task using a different subset of mechanosensitive afferent neurons. The reason why the number of trials in Fig 6F is smaller than the number of trials in Fig 6D is because all mice were tested for more than one amplitude. In Fig 6D, trial numbers reflect all behavioral trials including all amplitudes (although only the data from one amplitude, for each mouse, is shown in 6D and 6E). In Fig 6F, the raster plot shows the licking of a mouse example to a single type of amplitude. In other words, the 60 trials from Fig 6F come from a larger behavioral session where there were 60 trials of the analyzed amplitude, plus additional trials of other amplitudes that are not shown in the raster plot. As stated in the methods, for all mice, we tested amplitudes of 1.5 mN and 3mN. Since some mice had a perceptual deficit at the smaller amplitude and some mice had a deficit in both, we chose to quantify in Fig 6D and 6E the largest amplitude affected for each mouse. We have included new text in the materials section to clarify this.

7) Figure 4H: the authors performed paired t-tests for analysis of this dataset. A repeated measures, two-way ANOVA with post-hoc test would be more appropriate for this experiment (or at least some other analysis method that considers multiple comparisons). Is there no statistically significant difference in d' between 'after' and 'recovery' phases? If not, this needs to be stated and explained.

We have corrected figure 4H (now Figure 6H) and used two-way ANOVA with Bonferroni post-hoc for statistical analysis.

There is no statistically significant difference between the manipulation session (after, in the graph) and the recovery session, but there is between the manipulation session (after) and the control session (opposite paw control). A likely explanation for this incomplete recovery of performance could be that mice have long-lasting effects of the optogenetic inhibition. Another could be that mice experienced extinction learning during the manipulation session. In that session, mice found it more difficult to detect stimuli and obtain rewards, and this may alter the motivation levels of the mouse, which in turn could have slightly impacted the performance on the subsequent session. Given that the performance of the mice in the manipulation day was increasing over trials (Fig. 6D-F), we find it unlikely that mice still had mechanosensitivity altered 1 day after the manipulation and thus expect the incomplete recovery might be due to cognitive reasons, but to test this would require a great deal of new experiments and lead to a different study. Nevertheless, the Hit and False Alarm rates in Figure 6G show that most mice perform significantly on the recovery day, and later reach performance levels equivalent to pre-treatment levels on the opposite paw control session.

Reviewer #2 (Remarks to the Author):

The paper follows a previous publication (Abdo et al., Science 365, 695–699, 2019) from the same group that identified a novel peripheral glial cell subtype. The previous paper proposed the existence of specialized Schwann cells on the very terminal region of cutaneous primary sensory neurons that, with their extensive processes, signal thermal and mechanical noxious stimuli to the neurons. The paper further investigates with more detailed findings the different types of sensations that nociceptive Schwann cells convey to specific subsets of primary sensory neurons. A major observation is that nociceptive Schwann cells also surround the nerve-endings forming Meissner's corpuscle, thus being implicated in the perception of vibrotactile stimuli.

It would be helpful for readers if the authors better clarified that the original observation reported in the previous paper (Abdo et al., Science 365, 695–699, 2019), that nociceptive Schwann cells mediate both mechanical and thermal stimuli, is only partially confirmed in the present study, which shows that only mechanotransduction implicates a role of nociceptive Schwann cells.

In Abdo et al., subthreshold activation of nociceptive Schwann cells potentiated both thermal and mechanical stimuli. Because forced activation of nociceptive Schwann cells activates polymodal fibers (actually shown directly here in the present manuscript) then it is unsurprising that behavioral sensitivity to thermal stimuli is enhanced after polymodal C-fiber activation. It is, for example, well known that activation of polymodal nociceptors with electrical stimuli sets-up a central sensitization that underlies both mechanical and heat hyperalgesia. It should also be noted that in the experiments of Abdo et al that when nociceptive Schwann cells were silenced only mechanical-, but not heat-induced behavior was affected. We have introduced new text in the manuscript clarifying these points.

1) The authors should discuss the finding that half of C-M and A-M are non-responsive to blue-light. Are these non-responsive nerve fibers not surrounded by nociceptive Schwann cells? This question may also apply to the few non-responsive polymodal C-fibers, most RAM and all SAM. The findings seem to imply that two populations of DRG neurons exist, one with nociceptive Schwann cells on their terminal fibers and another without. If this is the case, what are the differences between the two populations in signaling innocuous and noxious stimuli?

This is a very interesting point and we have also addressed a similar question from reviewer 1 above. We believe that there are different degrees of connectivity between sensory Schwann cells and nociceptive neurons. This is obvious when one compares to different subtypes of nociceptors C-polymodals and C-mechanoreceptors. Almost all C-polymodals are driven by blue light but just 50% of C-Ms. Additionally, the C-M neurons are on average driven less efficiently (fewer APs over the 10 s of light stimulation. If the differences in connectivity's were a technical issue (see answer to Reviewer 1 above) then we would see on average the same physiological coupling regardless of afferent type. In answer to the reviewers second point we show quite clearly that the nature and type of coupling is very different between nociceptors and mechanoreceptors. We have now added data on mechanoreceptors that can be driven by optogenetic stimulation of Sox2 cells that are present only at the endings of SAM and RAM neurons. Indeed, we provide new evidence that RAMs are coupled to perhaps two separate populations of sensory Schwann cells.

One last point about the reliability of coupling is that there may be technical limitations that prevent seeing functional coupling in 100% of connected cells. For example, it could be that in some cases expression of the light sensitive protein is not optimal or that AP are initiated but extinguished at branch points (see above). Indeed, in the paper from Albers and collaborators they expressed Chr2 under the control of Peripherin promotor which should express in all sensory neurons (PMID: 26329459). They used a similar ex vivo preparation to that shown here to excite sensory endings directly with light (here there would be no functional coupling involved). Nevertheless, in that study only 67% (25/37) of C-fibers were directly excited by light. Thus, there may be a technical limit to record reliably optogenetic responses from at most between 67 and 90%.

2) The number of mice varies between 4 and 5. Are these numbers sufficient? And how was the sample size calculated as the numerosity varies across experiments?

The main behavioral experiments have an $n = 5$ (before-after manipulation, 1 day recovery and opposite paw control, Fig. 6). Only two control experiments (stimulator noise test, Ext. Fig. 8; and ArchT-negative mice, Ext. Fig.9) have a lower number of $n = 4$, where we expected a clear outcome (reduction of performance to chance level in Ext. Fig. 8E and an equally strong performance in Ext. Fig. 9C-D). We did not perform previous sample size calculations, but the number of mice used for behavior in this study is similar to our previous publications on forepaw somatosensation (Vestergaard et al., 2023, Schwaller et al., 2021, Paricio-Montesinos et al., 2020).

3) Line 86: a brief description of the method would help the readers.

We have added more details about the method as appropriate

4) The authors usually use $P <$, but sometimes (line 65, 199) they use $P =$. What is the reason for the difference? Sentences like ‘Schwann cells tended to show larger currents in response 66 to mechanical compared to blue light stimulation, but this not statistically different.’ should be avoided.

We have now changed this so that it is reformatted according to the journal’s guidelines.

We have omitted some of this data, as discussed above, so this sentence is deleted in the revised MS.

Minor:

5) SAM should be spelled out at its first appearance (line 163).

This has been corrected

6) Refs 5 and 7 are the same.

True, we have removed reference 7.

7) There are typos and unclear sentences. For example:

Line 653: a profound decreases

Line 213: We measured the force amplitude for the first spike as well as frequency following (where 1.0 denotes a spike evoked by every sinusoid) before and ten minutes after cyclical yellow light was focused on the receptive field

And others throughout the paper.

We have edited the offending text

Line 653: “ although, in the same fibers, profound decreases [...]”

Line 213: “We measured the force amplitude that triggered the first spike (threshold), as well as the percentage of subsequent sinusoids that triggered spiking (response frequency, where 1.0 equals one spike evoked by every sinusoid). We tested this before and after yellow light stimulation was applied to the receptive field.

Reviewer #3 (Remarks to the Author):

The detection of mechanical forces relies on the activation of sensory neurons that innervate the body and connect with the central nervous system. Somatosensory neurons are functionally and molecularly diverse with individual gene expression profiles tightly correlated with receptive tuning properties. The consensus view is that signal transduction occurs at the sensory neuron terminal. Consistent with this idea is the finding that sensory neurons express receptors that gated by environmental stimuli and these molecules are required for touch and temperature sensation.

It is also well established that sensory neuron endings are embedded within tissue where they form highly specialized endings and interact with other cells, including epithelial and glial cells. For example, in skin, Merkel cells are mechanosensory epithelial cells that contribute to touch responses. There is also evidence that keratinocytes play a role in mechanosensation. More recently, work from the Ernfors group has found that specialize glial cells called Terminal Schwann cells are required for normal sensory responses. This latter work led to the Ernfors group to postulate that terminal Schwann cells are bonafide sensory cells, challenging the idea of the passive/supportive roles typically assigned these cells.

While the notion that Schwann cells are principle detectors of mechanical stimuli is a very interesting hypothesis, it has remained unclear whether they have ‘instructive’ vs ‘permissive’ roles. Here, Ojeda-Alonso and colleagues have set-out to provide a causal link between terminal Schwann cell stimulation and sensory detection. They provide evidence that terminal Schwann cells are intrinsically mechanosensitive and that their direct stimulation is necessary and sufficient to drive sensory neuron firing in an ex vivo preparation. Lastly, they explore whether optically perturbing Schwann cell function alters perceptual thresholds.

Overall assessment:

This is undoubtedly an intriguing and potentially important area. However, there are several key experiments missing that are needed to substantiate this study’s conclusions. Specifically, better characterization of the mechanically evoked currents, validation of the inhibitory opsins effects on the Schwann cells and a positive control for the behavioral experiments are required. Lastly, the study presents an unbalanced view. The manuscript neglects to mention many well-established findings including the contributions of non-neuronal cells (for example, the work of Stucky, Lumpkin and Caterina Labs) and the fact that removing Piezo2 from sensory neurons completely blocks many aspects of touch perception (for example the work from Patapoutian, Chesler and Ginty labs). Furthermore, the references selectively neglect some groups while favoring others. The authors should provide a more scholarly and balanced view of the field. Last, the conclusions are out of proportion to the data. The effect sizes are relatively small in comparison to those seen with sensory ion channel knockouts or neuron ablation experiments. More measured and sober conclusions are warranted.

The reviewer makes four main points. The reviewer requests a better characterization of the mechanically activated currents. After looking critically at our data and making new attempts to patch isolated Sox10 cells in culture we have decided to remove this data from the manuscript. Sox10 cells in situ were shown by Abdo et al to wrap around the tiny ending of C-fiber and their morphology and physiological status in culture is very different from in vivo. They are very small spindle like cells that are difficult to patch and mechanical and reliably stimulate mechanically without compromising the electrophysiological recording. Indeed, these cells do not thrive at all well in the culture conditions we have tried. We have worked on this and have not yet found a better method to keep these cells in good condition, nor have we found conditions in which reliable patch

clamp experiments can be made. We do believe that these cells have a mechanosensitive current but it has been very hard to produce high quality data to be convincing for a published manuscript. This means that the second request of this reviewer is also not feasible to address at this time.

To the second point the original manuscript was written in a short format which precluded an extensive discussion of the literature. We have now included a more extensive comparison of our data with existing studies on the role of non-neuronal cells in mechanotransduction and believe that putting the present data in the context of other studies of non-neuronal sensory receptors has improved the manuscript. The major difference between our study and what we think makes our study stand out is that we provide evidence of direct coupling (perhaps electrical coupling) between sensory Schwann cells and nociceptors as well as RAMs. In the two studies that previously used optogenetic manipulations the latency of the responses were obviously quite long, 100's msec or seconds, but were actually not always quantified or measured. So in the case of Merkel cells the authors of the study from Lumpkin did not measure the latencies, but in example records they are clearly very long (Maksimovic et al PMID). In the case of the study of Albers and colleagues that looked at keratinocytes the vast majority of responses were seen in seconds not milliseconds. In our study more than 90% of responses were seen in much less than 1 second. We have introduced new text in the discussion regarding this.

The reviewer mentions the work of Ginty and Patapoutian on Piezo2. We are not clear of the direct relevance of this work and the data on mechanoreceptors from Piezo2 conditional knockouts that was actually collected in the one of the senior authors of the present manuscript. While it is clear that Piezo2 is contributing to sensory mechanotransduction, a number of issues remains to be clarified. For example, when Piezo2 is conditionally deleted in mice using Advillin-Cre ERT2 mice mechanosensitivity is only partly affected (DOI: 10.1038/nature13980). However, using HoxB8Cre mice as drivers, a much stronger phenotype is observed (DOI: 10.1126/scitranslmed.aat9897). This could be due to incomplete recombination as Advillin-CreERT2 marks 87% of primary sensory neurons DOI: 10.1038/nature13980. However, the HoxB8Cre line drives recombination in the developing spinal cord prior to neural crest delamination, and hence, any behavioral effect observed by conditionally deleting Piezo2 in HoxB8-lineage cells could arise from any progenies in this lineage, including sensory Schwann cells as

well as a more complete recombination in sensory neurons than Advillin. Thus, while it is clear that Piezo2 is essential for many aspects of touch, the exact contribution of different cell types such as Merkel cells, Meissner lamellar Schwann cells and nociceptive Schwann cells remains to be explored.

Finally, the reviewer requests that we use more measured language in our conclusions. This we have done in this new version of the manuscript.

Specific comments:

1) In Figure 1, the authors record from a small number of cells and only provide coarse information about the currents. The trace is low resolution. Shorter indentations (50-250msec) done in a stepwise series is need to better understand the kinetics and sensitivity of the evoked current. Given this is a 'new' type of mechano-current, the authors should show what is the voltage dependence, reversal, and ion permeability. Presumably this is a cation channel and excitatory?

We agree that it would be very interesting with further insights into the physiology of these cells and have tried to obtain such data. However, despite efforts we have had difficulties establishing high quality cultures enabling reliable characterization and have thus, not been able to include such data. See comments to Reviewer 1 and 2 and above. However, we have been able to add substantial new data both on a molecular and functional heterogeneity of sensory Schwann cells as well as a number of control experiments substantiating our results on both nociceptive Schwann cells and LTMR sensory Schwann cells. These data have been included in Figure 2, Figure 3C, Figure 4, Figure 5, Extended data Figures 1, 2, 3, 4, 5, 7, 9. We believe that these new experiments substantially strengthen the manuscript and hope that the reviewer finds the conclusions justified.

2) The authors should do current clamp recordings. Are Schwann cells excitable? The authors should demonstrate if they fire APs when stimulated. It seems like their resting potential is around -30 mV and their mechano-currents are small, so it is hard to understand how they might signal to an adjacent nerve ending. Similarly, it would be nice to see how ChR2 affects the membrane potential of these cells – are these cells excitable (possibly firing calcium spikes?) or would we just see a depolarization.

See comments to Reviewer 1 and 2 above

3) Much of the study relies on Arch for inhibition yet no characterization of this tool in Schwann cells is provided. It is unclear if/how Arch inhibits Schwann cells. The authors should perform some in vitro characterization of the effects of Arch on Schwann cell physiology. For example, it would be nice to see current clamp experiments showing if Arch hyperpolarizes the membrane and how long this takes. It is unclear how activation or inhibition of Schwann cells modulates the coupling between Schwann cells and nerve terminals. While dissecting out the nature of this coupling is well beyond the purview of this paper, some insight into the effects of the two opsins on Schwann cell membrane potential would help us understand how these manipulations affect the skin nerve prep recording or behavioural assays.

See general comments above. The main problem is that we do not think cells in vitro are healthy enough for such studies and furthermore, we question whether the dissociation of the glio-neural complex and analyzing only one component which morphologically and functionally is disrupted reflect the in vivo physiology. We have also discussed the mechanisms of inhibition in the Schwann cell and how this might indirectly affect the associated sensory ending (see revised Discussion). It should be noted that we have used the same optogenetic tool as others who have examined the role of non-neuronal cells in sensory mechanotransduction e.g. Lumpkin and colleagues.

4) The optical excitation of Schwann cells in the skin-nerve recordings is very nice. However, it would be better if the authors could provide a better idea of how tightly coupled things are. Does the spiking in the nerve correlate with the intensity, duration, and frequency of the optical stimulation of the Schwann cells? How repeatable is this (e.g. what if you stimulated 5 times in a row?). Does it fatigue during longer stimulation? Minor note- presumably the authors have made sure there is not any ectopic expression of ChR2 in sensory neurons due to low levels of Sox10 expression during development.

In most cases we used escalating intensities of light stimulation in order to determine the optimal light intensity to evoke action potentials. We used four light intensities and usually the second most intense stimulus already evoked a maximal response. This data is shown for all types of afferents examined and in general we saw very little sign of fatigue with the repeated stimuli as can be seen in the extended Figures 1C 2A (All nociceptors and thermoreceptors) and Extended Fig 4A and Extended Fig 7AP (All types of mechanoreceptors in hairy and glabrous skin). In addition to this extensive data, set we carried out a few experiments using high frequency pulses of blue light to excite RAMs in hairy skin. The three neurons examined could follow frequencies of up to 20 Hz

relatively well. But be consider this data preliminary and will not include it in the final MS.

5) In Figure 2I-L and similar, the summary data are highly processed. We would have more confidence in the authors conclusions if we could see the absolute numbers for spikes and threshold before and after illumination in the control and Arch mice. This should be represented with the individual replicates. This would allow the reader to better gauge the variability across units and assess for themselves the size and importance of the effects of Arch inhibition on stimulus evoked firing. The authors should explain in the main body of the text the criteria and procedure for including and excluding units in Figures 2J-L – it seems like the authors only quantified the units where Arch had an inhibitory effect, however the reader needs confidence this was done in an objective manner, to avoid biasing results. How did Arch affect the units that did not reduce their firing – did these units show no effect, or did they perhaps increase

their firing – extended data showing these units would be very helpful.

We are not hiding any data we made clear that in each case that an inhibition was considered to take place if threshold or firing decreased by more than 20%. Indeed we have plotted all units from this experiment individually in Extended data Fig 1J,K. It is very obvious from these plots that many units from Sox10-ArchT mice show dramatic changes in sensitivity whereas this rarely or never happens in control fibers.

6) The representative image in Fig 3 should be improved to more convincingly show this is a Meisner corpuscle and to better delineate the nerve endings and the Sox10. At least in the downloaded version the green is impossible to see, and the red is diffuse. The DAPI image does not add much.

The images in question have been replaced and a new extended data figure 4 added to show higher resolution images.

7) The behavioral data show some effect, but it is clear the mice can still detect the stimuli. The conclusions therefore need to be toned down quite a bit. Arch-negative mice should be included to control for the long light stimulation/paw restraint.

The behavioral effect is in fact robust and impressive, and the first time it has been shown that Schwann cells are necessary to set behavioural thresholds for touch. ArchT-negative mice have been tested as a control and this data is now shown as the Ext. Fig. 9 of the manuscript. Unlike Sox10-ArchT mice, ArchT-negative mice did not show impairment of vibrotactile detection impaired upon yellow light stimulation of the trained forepaw.

REVIEWER COMMENTS

Reviewer #1 (Remarks to the Author):

The authors have made substantial changes to the manuscript. This includes additional data to support the studies and substantial text revisions for more appropriate interpretations of the findings. I feel that my concerns have been addressed and the manuscript is suitable for publication.

Reviewer #2 (Remarks to the Author):

The Authors have satisfactorily answered to all the raised questions. I only add that a cartoon/graphical abstract that summarizes the major molecular and cellular findings might be of help to facilitate and widen the readership

Reviewer #3 (Remarks to the Author):

We provided what we hoped was a thorough, constructive and helpful review. We requested the authors address 3 major points regarding the lack of information about the mechanically evoked currents, the electrical properties of the Schwann cells and, related, what Arch activation was doing to these cells. These are crucial for interpreting the findings presented in the manuscript. The authors did not address these points in their revised manuscript and instead have removed one of their key findings. This diminishes the potential importance of the study: although the data showing perturbing Schwann cells alters mechanosensation, but not thermosensation is interesting, most of the other conclusions remain poorly supported by the data and ignore key findings in the literature (see Lehnert et al 2021; von Buchholtz et al 2021, Chirila et al 2022 regarding Piezo2 and sensory neurons in touch).

Reviewer #1 (Remarks to the Author):The authors have made substantial changes to the manuscript. This includes additional data to support the studies and substantial text revisions for more appropriate interpretations of the findings. I feel that my concerns have been addressed and the manuscript is suitable for publication.

We are very happy to read that this reviewer is fully satisfied with our revised manuscript. We thank the reviewer for his/her constructive comments

Reviewer #2 (Remarks to the Author):The Authors have satisfactorily answered to all the raised questions. I only add that a cartoon/graphical abstract that summarizes the major molecular and cellular findings might be of help to facilitate and widen the readership
We are very happy to read that this reviewer is fully satisfied with our revised manuscript. We thank the reviewer for his/her constructive comments. We have now included, as requested, a cartoon that summarizes the major findings (new Fig. 7).

Reviewer #3 (Remarks to the Author):

We provided what we hoped was a thorough, constructive and helpful review. We requested the authors address 3 major points regarding the lack of information about the mechanically evoked currents, the electrical properties of the Schwann cells and, related, what Arch activation was doing to these cells. These are crucial for interpreting the findings presented in the manuscript.

The reviewer(s) express the opinion that further characterization of sensory Schwann cells in culture is necessary to interpret the findings in the manuscript. On the contrary, we strongly believe that such experiments do not clarify, but rather lead to confusion and could even lead to gross misinterpretation of the in vivo data. We have already set out our reasoning in the previous rebuttal, but here again our main points:

Sox10-positive cells in culture are necessarily not the same things as Sox10+ve Schwann cells that enrap nociceptor endings or contact LTMRs in the in vivo setting. Indeed, it has been known for many decades that cultivated Schwann cells can rapidly change their phenotype and thus we have really no guarantee that what we record in vitro reflects the in vivo situation. Second the new data in the paper (especially the revised version) clearly illustrates the functional and anatomical diversity of the Sox10+ve Schwann cells in vivo. Therefore, it is impossible to say whether a Sox10+ve Schwann cell in culture had been connected within the Meissner's corpuscle to RAMs or had been connected to nociceptor endings. Indeed, our data clearly show that connectivity was functionally diverse even amongst different types of nociceptor (revised MS Figure 1).

We also take issue with the suggestion that we have ignored the reviewer's initial comments. For example, this/these reviewers also suggested we carry out further control experiments for the behavioural experiments. This extensive set of experiments was carried out and clearly showed that presence of ArchT in Sox10 cells was necessary to see the change in touch perception. (see revised Supplementary Fig. 9.

The authors did not address these points in their revised manuscript and instead have removed one of their key findings. This diminishes the potential importance of the study: although the data showing perturbing Schwann cells alters mechanosensation, but not thermosensation is interesting, most of the other conclusions remain poorly supported by the data and ignore key findings in the literature (see Lehnert et al 2021; von Buchholtz et al 2021, Chirila et al 2022 regarding Piezo2 and sensory neurons in touch).

We strongly disagree that the absence of the said data diminishes the importance of the study. For both technical and conceptual reasons, we were not convinced that leaving the data on

mechanosensitive currents in cultured Schwann cells helped the key message of the paper. The other two reviewers apparently also did not have any problem with this decision about what data we wished to publish. We do take exception to the rather bold and vague statement that “other conclusions remain poorly supported by the data”. It is not clear precisely what conclusions the reviewer(s) are referring to here. Finally, the reviewers seem to be upset that we did not refer to three papers in which Piezo2 mutant mice were examined. This is puzzling as the paper is not about Piezo2 and the relationship of those papers to the present paper is obscure. We mentioned Piezo2 precisely once in the manuscript in the context that Piezo2 deletion does not appreciably affect nociceptor mechanosensitivity. We quoted the original papers by Murthy et al in which the Lewin group made the physiological recordings. The three papers mentioned by the reviewer(s) do not have at their core the function of Piezo2 in sensory transduction, but rather use conditional Piezo2 mutants as tools in their studies. The three studies have in common that they claim that all LTMRs lose their mechanosensitivity in the absence of the Piezo2 gene. We have not ourselves observed such a dramatic loss of LTMR mechanosensitivity in conditional Piezo2 mutants (Murthy et al 2018) and we believe the differences between the studies are likely to be technical in nature. This particular controversy has nothing to do with the subject of this paper. We have nevertheless quoted the said papers in the context of the relative lack of effect of Piezo2 deletion on nociceptor mechanosensitivity.

REVIEWER COMMENTS

Reviewer #1 (Remarks to the Author):

While I do feel that the authors have made substantial changes and improvements to the manuscript, Reviewer 3 raised a few questions that I agree with and warrant additional consideration. In addition, higher resolution images in this round of submission suggest there are some questions regarding the expression of Sox10 in neurons that should be clarified.

- Given that the authors removed some of the in vitro data, I think it would be helpful to provide a short section in the discussion to explain why current cell culture experiments may not be optimal for characterization of specific Sox10+ Schwann cells (given the great diversity of Schwann cells etc.).
- Page 14, Line 411-415: "Conditional deletion of the Piezo2 gene using a Hox8b-Cre leads to loss of PIEZO2 in caudally located DRGs, but also in many skin cells which may include nociceptive Schwann cells. Nevertheless, nociceptors are largely unaffected in their mechanosensitivity in Hox8bCre:Piezo2 conditional mutant mice."
However, in Murthy et al., 2018, the authors conclude that 'mechanical stimuli-induced nociceptor firing is impaired in Piezo2HoxB8 mice' (see Figure 3). The wording in the present discussion seems inconsistent with the authors' prior interpretations of their data. (Also, mechanosensitivity is misspelled).
- From rebuttal: "We have nevertheless quoted the said papers in the context of the relative lack of effect of Piezo2 deletion on nociceptor mechanosensitivity." The authors do cite these three studies in the discussion, but the sentence does not describe the differences observed in experiments that involve Piezo2 deletion in sensory neurons across the different research groups.
- A more complete discussion of the Piezo2 literature is helpful for understanding the contributions of sensory neurons and end organs to stimulus transduction pathways. While Piezo2 is not discussed at length, the focus of the study is on mechanical nociception, which does involve Piezo2.
- In the new files, some images are higher resolution, and it looks like there may be Tomato expression in some of the neural processes (see Figure 3)?

Prior studies indicate there is "strong expression of Sox10 occurs throughout the peripheral nervous system during mouse embryonic development"
[see Pusch et al., 1998, in which the authors note Sox10 expression in DRG and spinal nerves; also see Bondurand et al., 1998: "SOX10 was strongly expressed in spinal nerves linked to dorsal root ganglia, and in dorsal root ganglia themselves (Fig. 3A, C)"].

What experiments have the authors performed to demonstrate that ChR2 is not being driven in some or all of the sensory neurons themselves? In the prior study in Science, there are places in Supplemental Figure S3 where Sox10TOM colocalizes with PGP9.5 or NF200 labeling.

Minor: In several places, Hoxb8 is written as Hox8b.

Reviewer #1 (Remarks to the Author):

While I do feel that the authors have made substantial changes and improvements to the manuscript, Reviewer 3 raised a few questions that I agree with and warrant additional consideration. In addition, higher resolution images in this round of submission suggest there are some questions regarding the expression of Sox10 in neurons that should be clarified.

We appreciate that the reviewer has taken time to help in clarifying any minor issues with what we hope will be the final version of our MS.

- Given that the authors removed some of the *in vitro* data, I think it would be helpful to provide a short section in the discussion to explain why current cell culture experiments may not be optimal for characterization of specific Sox10+ Schwann cells (given the great diversity of Schwann cells etc.).

Response: This point is well taken and we previously provided a section addressing this point as we have already extensively discussed with the reviewers of the manuscript. We have nevertheless elaborated further on this in the discussion section and it now reads:

“*In vivo* Sox10⁺ cells are both anatomically and functionally diverse and it is not presently possible to determine the original nature or origin of a cultured Sox10+ cell, limiting functional characterization of specific subtypes *in vitro*. Furthermore, numerically speaking nociceptive Schwann cells greatly outnumber the Sox10+ cells associated with mechanoreceptors.”

- Page 14, Line 411-415: “Conditional deletion of the Piezo2 gene using a HoxB8-Cre leads to loss of PIEZO2 in caudally located DRGs, but also in many skin cells which may include nociceptive Schwann cells. Nevertheless, nociceptors are largely unaffected in their mechanosensitivity in HoxB8Cre:Piezo2 conditional mutant mice.”

However, in Murthy et al., 2018, the authors conclude that ‘mechanical stimuli-induced nociceptor firing is impaired in Piezo2HoxB8 mice’ (see Figure 3). The wording in the present discussion seems inconsistent with the authors’ prior interpretations of their data.

Thank you for pointing this out, unfortunately we did oversimplify things in the original text. Indeed, the reviewer is correct that nociceptors in Piezo2 cKO mice do exhibit a change in their mechanosensitivity, albeit a mild change. Thus, we found in that study that the in the absence of Piezo2 the initial dynamic phase of nociceptor firing was blunted in both Adelta and C-fiber nociceptors, however we found no evidence that any nociceptors completely lost mechanosensitivity in the absence of Piezo2. We have reworded the sentence to make this clearer and it now reads:

“Mice with a HoxB8-Cre driven conditional deletion of Piezo2 show a substantial loss of mechanoreceptor function and A δ - and C-fiber nociceptors show blunted dynamic responses to noxious pressure⁴. However, there was no indication that A δ - and C-fiber nociceptors lost their mechanosensitivity in these mice.”

(Also, mechanosensitivity is misspelled).

This has been corrected.

- From rebuttal: “We have nevertheless quoted the said papers in the context of the relative lack of effect of Piezo2 deletion on nociceptor mechanosensitivity.” The authors do cite these three studies in the discussion, but the sentence does not describe the differences observed in experiments that involve Piezo2 deletion in sensory neurons across the different research groups.

- A more complete discussion of the Piezo2 literature is helpful for understanding the contributions of sensory neurons and end organs to stimulus transduction pathways. While Piezo2 is not discussed at length, the focus of the study is on mechanical nociception, which does involve Piezo2.

We have extended the discussion along the line proposed by the reviewer including differences observed in experiments involving Piezo2 deletion in sensory neurons as well as its contribution in sensory neurons and end organs to stimulus transduction pathways. Thus, to keep the text volume down, we merged these two discussion points into one new paragraph. The following paragraph has been added:

It is clear that Piezo2 is essential for sensory mechanotransduction in many mechanoreceptors^{4,6,47–50}. However, in two studies in which direct recordings were made from single mechanoreceptors it was found that many mechanoreceptors were still mechanosensitive in Piezo2 conditional knockout mice, in both cases Cre lines were used that drive recombination below cervical levels^{4,47}. Using the same conditional mutant mice as Hoffman et al, recordings from the DRG made with a multi electrode array claimed absence of mechanoreceptor activity⁵⁰ and recordings from dorsal horn neurons in the same conditional mutant mice also indicated loss of mechanoreceptor input⁴⁸. Similarly, using calcium imaging methods, it was claimed an almost complete absence of mechanoreceptor function in the cervical sensory ganglia of mice in which Piezo2 was conditionally deleted using a viral approach. Mechanosensitivity, is only present at the receptive field in the skin, with no direct involvement of the cell body. It seems likely that methods focused on recording from or imaging the cell bodies of sensory neurons underestimate the amount of intact transduction remaining after Piezo2 gene deletion. It is also possible that Piezo2 gene deletion or even loss of mechanosensitivity in mechanoreceptors could have indirect effects on primary afferent connectivity. Here we have not addressed the molecular nature of mechanotransduction in sensory Schwann cells. There is, however, evidence that Piezo1 may be a mechanotransducer in keratinocytes would could act to amplify mechanical nociception¹⁷, but the mechanisms appear distinct from the nociceptive Schwann cells studied here.

In vivo Sox10+ cells are both anatomically and functionally diverse and it is not presently possible to determine the original nature or origin of a cultured Sox10+ cell, limiting functional characterization of specific subtypes in vitro. Furthermore, numerically speaking nociceptive Schwann cells greatly outnumber the Sox10+ cells associated with mechanoreceptors. So far, we have little information about the molecular nature of mechanotransduction in Schwann cells associated with nociceptors or mechanoreceptors. Mice with a HoxB8-Cre driven conditional deletion of Piezo2 show a substantial loss of mechanoreceptor function and A δ and C-fiber nociceptors show blunted dynamic responses to noxious pressure⁴. However, there was no indication that A δ and C-fiber nociceptors lost their mechanosensitivity in these mice. It is likely that the Hoxb8 promoter drives recombination in Sox10+ Schwann cells which suggests that Piezo2 in Schwann cells does not substantially contribute to nociceptor transduction an idea that remains to be tested directly.

- In the new files, some images are higher resolution, and it looks like there may be Tomato expression in some of the neural processes (see Figure 3)?

We appreciate the concern of the reviewer. However, we are very confident this is not the case and have clarified this in the results section. The following text has been added:

“Sox10-TOM-labeled cells were also found within Meissner’s corpuscles (Fig. 3A,B) and hair follicles¹⁹. Consistent with Abdo, et al., 2019¹⁹, Sox10-TOM-labeled cells were not observed in any cutaneous afferents (Fig. 3A,B, Extended Data Fig. 3).”

Representative images in current manuscript and in Abdo et al. 2019 are maximum intensity projections of Z-stacks taken at 1 μ m intervals, so considering that nociceptive Schwann cells ensheath nerve endings it could appear that tomato is expressed in nerves due to the overlap that occurs when the layers are stacked.

In our experimental paradigm, we have taken into consideration that Sox10 is expressed in neural crest cells during development which will give rise to neurons and later on to glial cells of the PNS. However, we are using a Sox10CreERT2 driver mouse line. Thus, unlike a Cre driver, the recombination is subject to Tamoxifen-induced recombination. At postnatal stages when we induce recombination, Sox10 is only expressed in glial cells. Thus, this is why we have used a Sox10CreERT2 mice model, to drive expression in a time-dependent manner. Tamoxifen was injected in adult mice for all behavioral experiments. Considering the possibility of a leaking expression in neurons, in Abdo et al. 2019, to avoid the misperception that it could occur at the nerve endings we quantified tdTomato recombination in the cell bodies of DRG neurons. We did not find expression of Tdtomato in the cell bodies of DRG neurons (Abdo et al, 2019, Supplementary Fig 3C, 3D).

As an example, we show below one single plane of a Meissner corpuscle from the current experiments showing that PGP9.5 (in green) is observed where no tdTomato fluorescence is seen (upper arrows). At the nerve fiber entering the corpuscle (in green, lower arrow), we can see how the tdTomato expression is surrounding the nerve fiber. A full view of several optical sections is available in Extended Figure 3 in the manuscript.

Prior studies indicate there is “strong expression of Sox10 occurs throughout the peripheral nervous system during mouse embryonic development”

[see Pusch et al., 1998, in which the authors note Sox10 expression in DRG and spinal nerves; also see Bondurand et al., 1998: “SOX10 was strongly expressed in spinal nerves linked to dorsal root ganglia, and in dorsal root ganglia themselves (Fig. 3A, C)”].

We agree, but we use a conditional allele to temporally limit recombination and hence, there is not recombination during development. Please see response to the previous point.

What experiments have the authors performed to demonstrate that Chr2 is not being driven in some or all of the sensory neurons themselves? In the prior study in *Science*, there are places in Supplemental Figure S3 where Sox10TOM colocalizes with PGP9.5 or NF200 labeling.

We have quantified recombination in sensory neurons. This data is available in Fig S3 of Abdo et al., 2019 and pasted below (Tlx used as a marker of neurons and Sox10 as a marker of glial cells, note no recombination in neurons). Also, see response to the previous point. Regarding colocalization in nerves (which of course cannot happen if there is no recombination in the cell somas), these are small fibers with a thin ensheathment of glial processes and which in Z-stack projections might partially overlap, but when examining optical sections one can visually appreciate the subcellular localization of the Tomato and nerve markers. See response to the previous point.

Minor: In several places, Hoxb8 is written as Hox8b.

This has been corrected

Editorial Note: Panel above is Figure S3D from Abdo, H. et al., Specialized cutaneous Schwann cells initiate pain sensation. *Science (New York, N.Y.)* **365**, 695–699 (2019). Reprinted with permission from AAAS.